# MetaSym: A Symplectic Meta-learning Framework for Physical Intelligence

**Pranav Vaidhyanathan**\*  *pranav@robots.ox.ac.uk*
*Department of Engineering Science*
*University of Oxford, UK*

**Aristotelis Papatheodorou**\*  *aristotelis@robots.ox.ac.uk*
*Department of Engineering Science*
*University of Oxford, UK*

**Mark T. Mitchison**  *mark.mitchison@kcl.ac.uk*
*School of Physics, Trinity College Dublin, Ireland*
*Department of Physics, King's College London, UK*

**Natalia Ares**  *natalia.ares@eng.ox.ac.uk*
*Department of Engineering Science*
*University of Oxford, UK*

**Ioannis Havoutis**  *ioannis@robots.ox.ac.uk*
*Department of Engineering Science*
*University of Oxford, UK*

**Reviewed on OpenReview:** *https://openreview.net/forum?id=MV1wfMe647*

## Abstract

Scalable and generalizable physics-aware deep learning has long been considered a significant challenge with various applications across diverse domains ranging from robotics to molecular dynamics. Central to almost all physical systems are symplectic forms, the geometric backbone that underpins fundamental invariants like energy and momentum. In this work, we introduce a novel deep-learning framework, MetaSym. Our approach combines a strong symplectic inductive bias obtained from a symplectic encoder, and an autoregressive decoder with meta-attention. This principled design ensures that core physical invariants remain intact, while allowing flexible, data efficient adaptation to system heterogeneities. We benchmark MetaSym with highly varied and realistic datasets, such as a high-dimensional spring-mesh system (Otness et al., 2021), an quantum system with dissipation and measurement backaction, and robotics-inspired quadrotor dynamics. Crucially, we fine-tune and deploy MetaSym on real-world quadrotor data, demonstrating robustness to sensor noise and real-world uncertainty. Across all tasks, MetaSym achieves superior few-shot adaptation and outperforms larger state-of-the-art (SOTA) models.

## 1 Introduction

Learning to predict the dynamics of physical systems is a fundamental challenge in scientific machine learning, with applications ranging from robotics, control, climate science, and quantum computing (Ghadami & Epureanu, 2022; Zhang et al., 2024; Alexeev et al., 2024). Traditional approaches often rely on carefully derived differential equations (Papatheodorou et al., 2024) that embed known conservation laws and geometric properties (e.g., Hamiltonian or Lagrangian mechanics). Although these classical models have

---

\*Equal Contribution

been tremendously successful in capturing fundamental physics, they can become unwieldy or intractable when faced with complex real-world phenomena, such as high-dimensional systems, intricate interactions, or partially known forces, that defy simple closed-form representations. Recent progress in deep learning has opened new avenues for data-driven modeling of dynamical systems, bypassing the need for complete analytical descriptions (Carleo et al., 2019). Neural ODE frameworks such as Chen et al. (2019), for instance, reinterpret dynamic evolution as a continuous function learned by a neural network, while operator-learning approaches such as Fourier Neural Operators (FNOs) in Li et al. (2021) allow for flexible mappings from initial conditions to solutions of partial differential equations. Despite these advances, deep learning approaches often face two critical challenges:

- **Preserving the underlying physical structure**. Standard networks, left unconstrained, may inadvertently violate symplectic forms, conservation of energy, or other geometric constraints intrinsic to physical dynamics (Chen et al., 2020). These violations can accumulate over time, producing qualitatively incorrect long-horizon predictions, e.g. spurious energy drift in Hamiltonian systems.

- **Generalizing across related systems**. Many real-world applications involve entire families of similar yet distinct systems (e.g., variations of a robotic manipulator differing in load mass or joint friction, or molecular systems differing in exact bonding parameters). Training an entirely separate model for each variant is both data-inefficient and computationally expensive. Without mechanisms to share knowledge, a network trained on one system will typically fail to adapt efficiently to another, even if they exhibit similar physical behaviors. Bridging this sim-2-real gap is crucial for a variety of tasks such as control and real-time prediction Bai et al. (2024).

The trade-off between flexibility (i.e., capacity to learn diverse dynamics) and the enforcement of physical constraints can be addressed through specialized architectures that embed geometric priors (Chen et al., 2020; Greydanus et al., 2019) related to the physical system. Notably in the context of Hamiltonian mechanics, the *Symplectic Networks* (SympNets), introduced in Jin et al. (2020), incorporate symplectic forms directly into their design, guaranteeing that learned transformations represent canonical transformations. This preserves key invariants such as energy and momentum, mitigating error accumulation in long-horizon forecasts.

However, real-world systems also exhibit heterogeneity, i.e., varying parameters, boundary conditions, or even control signals that deviate from conservative dynamics. Thus, *meta-learning* becomes a natural extension (Schorling et al., 2025). By training on a set of related systems, meta-learning-based methods (e.g., Model-Agnostic Meta-Learning (MAML), interpretable Meta neural Ordinary Differential Equation (iMODE) and Fast Context Adaptation Via Meta-Learning (CAVIA) (Finn et al., 2017; Li et al., 2023; Zintgraf et al., 2019b)) acquire high-level inductive biases that can be quickly adapted to novel systems using limited additional data. Consequently, when one faces a new variant of a familiar system, the network can fine-tune a handful of parameters, rather than retrain from scratch. This provides robust and scalable performance.

## 1.1 Contributions

In this work, we introduce **MetaSym**, a deep-learning framework that addresses the major challenges of data-driven modeling of physical systems, i.e., preserving underlying geometric structures to ensure physically consistent behavior over long time-horizons and rapidly adapting to system variations with minimal data. Our contributions are listed below:

- **Symplectic encoder for structure preservation**. We propose a specialized neural encoder (SymplecticEncoder) built on SympNet modules. The inherent structural invariants of the SympNets provide a strong inductive bias to the SymplecticEncoder, while our bi-directional training pipeline enforces Hamiltonian consistency, obtained by the canonical transformations that pertain to different systems. Hence, the SymplecticEncoder's output conserves key geometric invariants (e.g., energy and momentum), effectively minimizing error accumulation over long-term roll-outs with minimal architecture size.

- **Autoregressive decoder with meta-attention for adaptation**. To handle non-conservative forces and variations in system parameters, we introduce *ActiveDecoder*, a transformer-inspired de-

coder module equipped with a meta-attention framework. This decoder incorporates control inputs, external forces, and per-system parameters enabling flexible modeling of real-world effects beyond ideal Hamiltonian dynamics, while enabling autoregressive multi-step prediction during inference time.

- **Meta-learning for multi-system generalization**. We adopt different meta-learning paradigms for the *SymplecticEncoder* and *ActiveDecoder* of our architecture motivated by specific design choices explained in Section 3. This yields a single framework that quickly adapts to new or modified systems, even in **real-world** scenarios.

By integrating physical constraints and generalizable architectural changes into a novel training pipeline that utilizes meta-learning updates for both our SymplecticEncoder and *ActiveDecoder*, we benchmark this bespoke smaller architecture against other SOTA deep learning methods, including Dissipative Hamiltonian Neural Networks (DHNNs) (Sosanya & Greydanus, 2022) and Transformers (Vaswani, 2017; Geneva & Zabaras, 2022), for modeling various physical systems in both classical and quantum regimes. These systems include a high-dimensional spring-mesh system, an open-quantum system whose dynamics are highly complex and counterintuitive in the classical regime, and a quadrotor with floating-base dynamics. This provides evidence of far reaching implications for a diverse set of physics modeling tasks including challenging tasks like *real-time* quantum-dynamics prediction and simulating a variety of complex dynamics that typically require complex computational methods to solve. To the best of our knowledge, MetaSym is the first bespoke physics-based deep-learning model to adapt and generalize well to both classical and non-unitary quantum dynamics.

## 2 Related Work and Background

Physicists have long utilized the Lagrangian and Hamiltonian formalisms of mechanics to study the dynamics of physical systems (Kibble & Berkshire, 2004). Consider $\mathbf{q} = (q_1, \ldots, q_d)$ that represents the generalized coordinates of an $d$-dimensional configuration space (position), while $\mathbf{p}$ represents the corresponding generalized momenta. We can describe the Hamiltonian $H(\mathbf{q}, \mathbf{p}, t)$ as the total energy of the system. This leads to Hamilton's equations as follows:

$$\dot{q}_i = \frac{\partial H}{\partial p_i}, \quad \dot{p}_i = -\frac{\partial H}{\partial q_i}, \quad i = 1, \ldots, d. \tag{1}$$

This formalism naturally imbues the phase space $(\mathbf{q}, \mathbf{p}) \in \mathbb{R}^{2d}$ with geometric structures that are vital to the study of these physical systems. One such structure is the symplectic form $(\omega)$ given by, $\omega = \sum_{i=1}^{n} \mathrm{d}p_i \wedge \mathrm{d}q_i$. Concretely, a map $\Phi_t : (\mathbf{q}(0), \mathbf{p}(0)) \mapsto (\mathbf{q}(t), \mathbf{p}(t))$ is said to be symplectic if $\Phi_t^* \omega = \omega$. This implies that the flow in the phase space preserves the volume (Royer, 1991).

Inspired by such geometric formulations of mechanics, recent work such as Hamiltonian Neural Networks (HNNs) and Lagrangian Neural Networks (LNNs) have sought to embed physical priors into deep-learning architectures (Greydanus et al., 2019; Cranmer et al., 2020). In particular, SympNets have emerged as structure-preserving neural architectures designed specifically for learning Hamiltonian systems, ensuring that their learned maps are intrinsically symplectic. These architectures admit universal approximation theorems and crucially, this construction does not require solving ODEs or differentiating a candidate Hamiltonian during training and inference times, which often leads to more efficient optimization compared to other architectures such as HNNs or LNNs. The collection of all SympNets forms a group under composition, ensuring that every learned map is invertible with a closed-form inverse. However, due to the fundamental nature of Hamiltonian systems and symplectic forms, SympNets and similar architectures fail to generalize to dissipative systems where the symplectic structure is no longer preserved (Chen et al., 2020; Cranmer et al., 2020; Jin et al., 2020). While there have been attempts to reconcile dissipative dynamics and control inputs to model realistic physical systems by preserving the symplectic framework, such as the Dissipative SymODEN (Zhong et al., 2020; 2024), they often suffer from the lack of generalization to different systems (Okamoto & Kojima, 2024).

Generalization between different but related physical systems is also vital for deep-learning methods to excel at physics modeling. Meta-learning serves as the natural avenue for exploring such strategies. Building on a series of meta-learning strategies (Finn et al., 2017; Rajeswaran et al., 2019; Zintgraf et al., 2019a; Nichol, 2018), the iMODE framework represents a notable advancement in applying deep learning to families of dynamical systems. In this setup, one set of parameters captures the universal dynamics shared across all systems in a family, while another set encodes the idiosyncratic physical parameters that differentiate one system instance from another. However, certain drawbacks still persist. Apart from the lack of physics priors, the existing meta-learning approaches, such as iMODE, suffer from a lack of scalability (Choe et al., 2023).

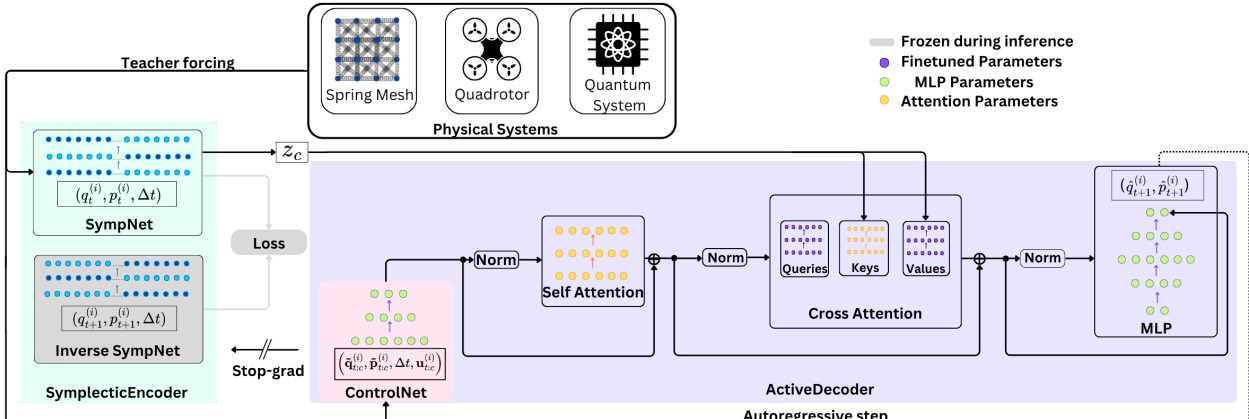

Figure 1: MetaSym integrates a *SymplecticEncoder* (light-green), *ActiveDecoder* (light-purple), and *ControlNet* (pink). The *SymplecticEncoder* is pre-trained in isolation on conservative state-space data using shared forward/inverse networks receiving $(\mathbf{q}_t^{(i)}, \mathbf{p}_t^{(i)}, \Delta t)$ and $(\mathbf{q}_{t+1}^{(i)}, \mathbf{p}_{t+1}^{(i)}, \Delta t)$ respectively, and thus explicitly enforcing time-reversibility. Subsequently, with the *SymplecticEncoder* frozen, the *ActiveDecoder* and *ControlNet* are then trained autoregressively via teacher forcing, where system-specific adaptation is achieved by fine-tuning the cross-attention's query/value parameters (purple dots in cross attention) with few-shot gradient steps. During inference, the *ControlNet* processes a sequence of non-conservative coordinates and control signals $\{\tilde{\mathbf{q}}_{t:T}^{(i)}, \tilde{\mathbf{p}}_{t:T}^{(i)}, \Delta t, \tilde{\mathbf{u}}_{t:T}^{(i)}\}$, while the *SymplecticEncoder* projects them onto the conservative manifold and integrates them on it, producing $\mathbf{z}_c$. The *ActiveDecoder* using its cross-attention, perturbs $\mathbf{z}_c$ to predict the dynamics of the system for future time-steps, autoregressively. We indicate the next-step predictions of our network with $\left(\hat{\mathbf{q}}_{t+1}^{(i)}, \hat{\mathbf{p}}_{t+1}^{(i)}\right)$.

## 3 Methods

In this section, we detail our pipeline for learning, adapting, and predicting the dynamics of physical systems using MetaSym, our structure-preserving neural architecture and meta-learning framework. We describe both the high-level design of our *encoder–decoder* model and the specialized training procedures implemented via *meta-learning*. Fig. 1 represents the overview of the architecture. Details on the described training and meta-learning pipelines are given in Appendix A, while each design choice is based on ablation studies provided in Appendix D.

### 3.1 SymplecticEncoder: Structure Preservation

To ensure that predictions preserve fundamental geometric invariants, the encoder module is implemented as a *SymplecticEncoder*. Internally, it uses a *symplectic neural network* (e.g., `LASympNet` by Jin et al. (2020)) consisting of sub-layers that update position $\mathbf{q}_t^{(i)}$ and momentum $\mathbf{p}_t^{(i)}$ for systems $i = 1, ..., n$ in a manner designed to approximate Hamiltonian flows and preserve symplecticity. Specifically, each sub-layer either

performs an "*up*" or "*low*" transformation,

$$
\begin{aligned}
\text{(Up) } \mathbf{q}_t^{(i)} &\leftarrow \mathbf{q}_t^{(i)} \alpha(\mathbf{p}_t^{(i)}) \Delta t, \quad \mathbf{p}_t^{(i)} \leftarrow \mathbf{p}_t^{(i)}, \\
\text{(Low) } \mathbf{q}_t^{(i)} &\leftarrow \mathbf{q}_t^{(i)}, \quad \mathbf{p}_t^{(i)} \leftarrow \mathbf{p}_t^{(i)} + \beta(\mathbf{q}_t^{(i)}) \Delta t.
\end{aligned}
\tag{2}
$$

Here, we use $\alpha$ and $\beta$ that can be either *linear* or *activation* modules, representing learnable parameters, ensuring we remain in the class of canonical (i.e., symplectic) maps. By stacking multiple up/low blocks, we achieve a deep network $\Phi_\theta : (\mathbf{q_t^{(i)}}, \mathbf{p_t^{(i)}}, \Delta t) \mapsto (\mathbf{q_{t+1}^{(i)}}, \mathbf{p_{t+1}^{(i)}})$. A crucial mathematical property is that the composition of symplectic transformations remains symplectic. Thus, when these modules are stacked together in sequence, regardless of network depth or configuration, the resulting transformation is guaranteed to be symplectic. This relies on the algebraic properties of the "Up" and "Low" updates, which act as symplectic shear transformations. For instance, the Jacobian matrix $\mathbf{J}_{low}$ of a "Low" update $(\mathbf{q}, \mathbf{p}) \mapsto (\mathbf{q}, \mathbf{p} + \beta(\mathbf{q})\Delta t)$ is lower-triangular with identity diagonals:

$$
\mathbf{J}_{low} = \begin{pmatrix} \mathbf{I} & \mathbf{0} \\ \frac{\partial \beta}{\partial \mathbf{q}} \Delta t & \mathbf{I} \end{pmatrix}.
\tag{3}
$$

For a map to be symplectic, its Jacobian must satisfy $\mathbf{J}^T \mathbf{\Omega} \mathbf{J} = \mathbf{\Omega}$. This condition holds for triangular maps provided the cross-term block (here, $\frac{\partial \beta}{\partial \mathbf{q}}$) is a symmetric matrix. To ensure this, we implement $\alpha$ and $\beta$ either as gradients of a scalar potential or using layers with symmetric weights (Jin et al., 2020).

A pivotal feature of Hamiltonian dynamics is *time reversibility*, i.e., if we integrate the system forward from $(\mathbf{q_t^{(i)}}, \mathbf{p_t^{(i)}})$ to $(\mathbf{q_{t+1}^{(i)}}, \mathbf{p_{t+1}^{(i)}})$ over time $\Delta t$, then integrating backwards over $-\Delta t$ should return the system exactly to $(\mathbf{q_t^{(i)}}, \mathbf{p_t^{(i)}})$. This property lies at the heart of many physical invariants (energy, momentum, etc.) and is crucial for long-horizon stability in forecasting the behavior of such systems.

To replicate this in the neural network, each forward pass provides the update for the forward and inverse network. The forward network is characterized as $\Phi_{\theta_{SE}}(\mathbf{q_t^{(i)}}, \mathbf{p_t^{(i)}}, \Delta t) = [\mathbf{q_{t+1}^{(i)}}, \mathbf{p_{t+1}^{(i)}}]$, and the inverse network $\Phi_{\theta_{SE}}^{-1}$ acts as not merely the computationally reversed pass of $\Phi_{\theta_{SE}}$, but also switches the sign of $\Delta t$ (i.e., steps "backwards in time"). From a physical standpoint, preserving both a forward and inverse map enforces the time-reversal symmetry characterizing Hamiltonian flows. We train a single model in both directions simultaneously by sharing gradients between the forward and inverse instances, effectively minimizing reconstruction errors. As a result, the network is less susceptible to artificial energy drift and can better maintain conservation laws over extended forecasts. More specifically, this bi-directional training pipeline acts as implicit regularization by enhancing the model's ability to faithfully approximate the underlying canonical transformations, while allowing consistent backward integration without introducing extraneous numerical artifacts. However, note that only the forward instance, $\Phi_{\theta_{SE}}$, is used during inference.

During training the encoder is provided with a sequence of phase-space points (T time-steps) and encodes them onto a conservative latent space $\mathbf{z}_c = [\mathbf{q_{enc}}, \mathbf{p_{enc}}] \in \mathbb{R}^{2d \times T}$ that is ultimately used by the *ActiveDecoder* in the cross attention mechanism to specifically fine-tune the query and value parameters.

Additionally, to ensure that our encoder can generalize quickly across multiple related but distinct physical systems, we adopt a MAML-style (Finn et al., 2017) framework. This meta-learning paradigm works particularly well in the *SymplecticEncoder's* case, since SympNets represent canonical transformations that are known to have well-defined Hessian (Birtea et al., 2020), which is crucial for methods like MAML that use second-order gradients. For each system $i$ in a mini-batch, we split its trajectory into $\mathcal{I}_{adapt}$ and $\mathcal{I}_{meta}$ sets. During the fast adaptation loop, we optimize the parameters of the encoder $\theta_{SE}$ *only* on $\mathcal{I}_{adapt}$, by minimizing the mean-squared error between its forward predictions and the ground-truth labels. This process simulates "specializing" the encoder to system $i$'s local dynamics using a simple loss function that avoids gradient-terms that may destabilize the subsequent meta-update of $\theta_{SE}$. For the outer loop, we perform a forward and an inverse pass of the *SymplecticEncoder* and subsequently we minimize a combined loss

represented as,

$$\mathcal{L}_{\text{meta}} = \frac{1}{\mathcal{T}_{meta} N_{batch}} \sum_{\substack{t \in \mathcal{I}_{\text{meta}} \\ i \in N_{batch}}} \left\| \Phi_{\theta_{SE}}\big(\mathbf{q}_t^{(i)}, \mathbf{p}_t^{(i)}; \boldsymbol{\theta}_{\boldsymbol{SE}}^{*(i)}\big) - \Phi_{\theta_{SE}}^{-1}\big(\mathbf{q}_{t+1}^{(i)}, \mathbf{p}_{t+1}^{(i)}; \boldsymbol{\theta}_{\boldsymbol{SE}}^{*(i)}\big) - [\mathbf{q}_{t+1}^{(i)} - \mathbf{q}_t^{(i)}, \mathbf{p}_{t+1}^{(i)} - \mathbf{p}_t^{(i)}] \right\|^2,$$
(4)

where $\mathcal{T}_{meta}$ is the total number of time-steps stored in $\mathcal{I}_{meta}$ set and $N_{batch}$ the total number of meta-learned systems in the batch. While the *SymplecticEncoder* is trained in isolation, we freeze its parameters during the *ActiveDecoder's* training.

## 3.2 ActiveDecoder: Autoregressive Decoder

As mentioned in Section 1.1, we train an autoregressive decoder to model non-conservative and realistic forces such friction, or air resistance, that break pure Hamiltonian symmetries. Unlike the *SymplecticEncoder*, which strictly enforces canonical updates, the ActiveDecoder can incorporate these extraneous phenomena.

We define $\tilde{\mathbf{q}}_{\mathbf{t}}^{(\mathbf{i})}$ and $\tilde{\mathbf{p}}_{\mathbf{t}}^{(\mathbf{i})}$ as the dissipative canonical coordinates, of a system $i$. An input sequence $\left( \tilde{\mathbf{q}}_{\mathbf{t:c}}^{(\mathbf{i})}, \tilde{\mathbf{p}}_{\mathbf{t:c}}^{(\mathbf{i})}, \Delta t, \mathbf{u}_{\mathbf{t:c}}^{(\mathbf{i})} \right) \in \mathbb{R}^{(2d+m+1) \times c}$ with a context-window of $c$ time-steps is provided to a linear projection, where $\mathbf{u}_{\mathbf{t}}^{(\mathbf{i})} \in \mathbb{R}^m$ represents the $m$-dimensional control-input driving the system at time $t$. This linear layer, which we call *ControlNet* as part of the *ActiveDecoder* serves to map the input to a latent vector $\mathbf{z}_d \in \mathbb{R}^{2d}$ representing the non-conservative part of the system's dynamics. We then apply a masked multi-head self-attention over the sequence $\mathbf{z}_d$ for autoregressive decoding. The masking allows us to model the causal dependencies of the decoder's input sequence.

Next, we apply a cross-attention mechanism, augmented with a specialized *meta-attention* design. To achieve multi-system generalization and fast adaptation, we use a meta-learning setup akin to a *bi-level* optimization inspired by recent progress (Li et al., 2025). We separate the ActiveDecoder's parameters into global parameters, $\theta_{AD}$ that remain *shared* across all systems, and local parameters, $\zeta_i$ that can be interpreted as system specific parameters. More specifically, in the meta-attention:

- **Key parameters** remain global and are updated in the outer loop, common to all systems.

- **Query/Value parameters** are re-initialized for each system and fine-tuned with a few iterations in the inner loop, allowing the decoder to discover per-system or per-task representations.

For each system $i$, we split the time-sequenced data, $\tilde{\mathbf{q}}^{(\mathbf{i})}, \tilde{\mathbf{p}}^{(\mathbf{i})}, \mathbf{u}^{(\mathbf{i})}$ into an *adaptation* set, $\mathcal{I}_{adapt}$ containing $\mathcal{T}_{adapt}$ number of time-steps, and a *meta* set $\mathcal{I}_{meta}$ with $\mathcal{T}_{meta}$ number of time-steps. During the *inner loop* of the bi-level optimization, we hold $\theta_{AD}$ fixed and fine-tune $\zeta_i$ by performing a few gradient steps. Each step iterates over $\mathcal{I}_{adapt}$ feeding the ActiveDecoder ($\Phi_{AD}$) with ground-truth $(\mathbf{q}_{t:c}^{(i)}, \mathbf{p}_{t:c}^{(i)})$ and control $\mathbf{u}_{t:c}^{(i)}$, and minimizing an inner MSE loss across $\mathcal{T}_{adapt}$ number of time-steps, i.e.,

$$\mathcal{L}_{\text{inner}}^{(i)} = \frac{1}{\mathcal{T}_{adapt}} \sum_{t \in \mathcal{I}_{\text{adapt}}} \left\| \Phi_{\theta_{\text{AD}}}\big(\mathbf{q}_{t:c}^{(i)}, \mathbf{p}_{t:c}^{(i)}, \mathbf{u}_{t:c}^{(i)}; \boldsymbol{\theta}_{\boldsymbol{AD}}, \boldsymbol{\zeta}_i\big) - [\mathbf{q}_{t+1}^{(i)}, \mathbf{p}_{t+1}^{(i)}] \right\|^2.$$
(5)

This ensures that $\zeta_i$ adapts to system or task specific idiosyncrasies.

Once the local parameters $\zeta_i$ have been adapted to the optimal value $\zeta_i^*$, we evaluate the model's performance on $\mathcal{I}_{meta}$ portion of the trajectory. The corresponding outer loss,

$$\mathcal{L}_{\text{outer}}^{(i)} = \frac{1}{\mathcal{T}_{meta}} \sum_{t \in \mathcal{I}_{\text{meta}}} \left\| \Phi_{\theta_{\text{AD}}}\big(\mathbf{q}_{t:c}^{(i)}, \mathbf{p}_{t:c}^{(i)}, \mathbf{u}_{t:c}^{(i)}; \boldsymbol{\theta}_{\boldsymbol{AD}}, \boldsymbol{\zeta}_i^*\big) - [\mathbf{q}_{t+1}^{(i)}, \mathbf{p}_{t+1}^{(i)}] \right\|^2,$$
(6)

propagates gradients *only* to $\theta_{AD}$. The overall training objective is then the sum over $\mathcal{L}_{\text{outer}}^{(i)}$ for all systems $i$, over the batch. We finalize the decoder with a Multi-Layer Perceptron (MLP) that outputs the position and momentum for the next time-step $\left( \hat{\mathbf{q}}_{\mathbf{t+1}}^{(\mathbf{i})}, \hat{\mathbf{p}}_{\mathbf{t+1}}^{(\mathbf{i})} \right)$. The architecture is then used autoregressively for future

predictions. The inherent high dimensionality of the ActiveDecoder's internal representations, coupled with the lack of well-defined or tractable Hessian guarantees in this setting (LeCun et al., 2012), motivates our adoption of this meta-learning paradigm over traditional methods such as MAML. By selectively adapting only a carefully chosen subset of *ActiveDecoder's* parameters, we are able to minimize both the number of adaptation steps and the quantity of data required during the meta-learning process.

### 3.3 Autoregressive Inference

Once both the *SymplecticEncoder* and *ActiveDecoder* (Sections 3.1 and 3.2) are trained, we generate future predictions by rolling out the model *autoregressively* during inference time. Specifically, at test time, we are given initial phase-space measurements. We first map to the conservative portion of the phase-space, $(\mathbf{q}_t^{(i)}, \mathbf{p}_t^{(i)})$ through the *SymplecticEncoder*. The *ActiveDecoder* then produces the next predicted phase-space trajectory $(\mathbf{\hat{q}_{t+1}^{(i)}}, \mathbf{\hat{p}_{t+1}^{(i)}})$, based on the pipeline highlighted in Section 3.2. However, we subsequently treat $(\mathbf{\hat{q}_{t+1}^{(i)}}, \mathbf{\hat{p}_{t+1}^{(i)}})$ as inputs for the next time-steps. This is repeated over the context window provided to the decoder.

In summary, by pairing our *SymplecticEncoder* (to ensure structure preservation) that provides a strong **inductive bias** with an *Autoregressive Transformer-style Decoder* (*ActiveDecoder*) equipped with *meta-attention* (to handle individual system variations), MetaSym can rapidly adapt to diverse physical systems while guaranteeing physically consistent core dynamics. The proof of this strong inductive bias provided by the *SymplecticEncoder* and its effect on the training of the *ActiveDecoder* as part of MetaSym is provided by Appendix C.

## 4 Results

In this section, we evaluate MetaSym's performance, against a variety of SOTA models and benchmarks, especially in long-horizon stability and few-shot generalization. In particular, we report the behavior of MetaSym against Transformers that have achieved impressive performance in modeling physical systems (Geneva & Zabaras, 2022) and DHNNs (Sosanya & Greydanus, 2022) that model systems via the Helmholtz decomposition, although they require gradient information and a numerical integrator to obtain trajectory data. Our model predicts the subsequent time-steps directly, which poses a much harder task that does not require numerical integration, which can be time-consuming and may hinder real-time performance. These results also reflect on the effectiveness of our design choices.

As mentioned in Section 1.1, we choose to benchmark MetaSym for three extremely challenging and diverse datasets. Notably, we choose the spring-mesh system, which tests scalability and closely resembles finite element modeling of surfaces with different materials. This dataset has provided the SOTA testbed for large dimensional systems (Otness et al., 2021). Moreover, we utilize a well proven, reproducible dataset of an open-quantum system undergoing heterodyne detection leading to representative quantum effects, such as decoherence, to demonstrate MetaSym's robustness to noise and flexibility to model all types of physical systems. Finally, we choose to predict the dynamics of a quadrotor, a long-standing benchmark in robotics and challenging due to floating-base dynamics and sensor noise that is simulated using standard Gaussian additive noise (Bansal et al., 2016). Furthermore, we also demonstrate the adaptability of MetaSym to real-world data by adapting this model (Mohajerin & Waslander, 2018). The description of each experimental setup is laid in the next sections, however a more extensive description can be found in Appendix B. All reported MSE errors are not weighted. In the plots, the errors correspond to each phase-space coordinate, while in Table 1 the reported errors are calculated across all phase-space dimensions. Finally, information regarding the utilized compute resources and training hyperparameters are provided in Appendices F & G.

### 4.1 Spring-mesh System

As our first benchmark, we use the 400-dimensional spring-mesh system introduced by Otness et al. (2021), consisting of a $10 \times 10$ lattice of point masses connected by elastic springs. Each of the 100 masses possesses two positional and two momentum degrees of freedom, yielding a 400-dimensional state space. This system,

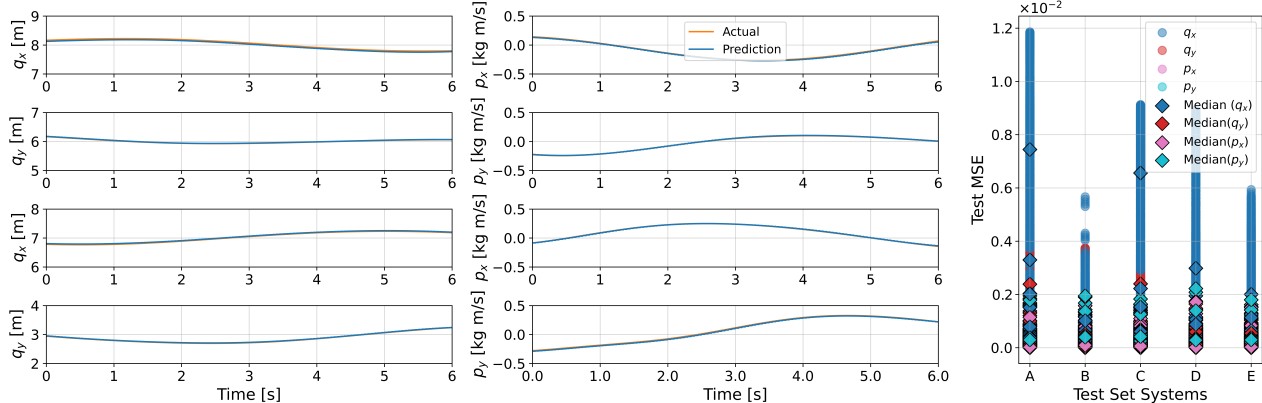

Figure 2: (**Left**) Time evolution of the system's position and momentum variables for a representative set of masses in the spring mesh. The orange curves represent the ground-truth trajectories for each phase-space coordinate type (i.e., $q_x, q_y, p_x$, and $p_y$), while the blue curves depict the model's predictions. The close alignment between these trajectories underscores the model's capacity to accurately capture the underlying long-term dynamics (600 time-steps) of the coupled spring system using a context window of 30 time-steps. **(Right)** Plots illustrating the mean squared error (MSE) of the time evolution of **each phase-space coordinate type** (dots) for five (A-E) spring-mesh systems in the test set. Each column encapsulates the spread of errors observed across all masses in the spring-mesh for a given phase-space coordinate across multiple time-steps, with the boxes marking their median values. The uniformly low median errors across all components demonstrate that the model generalizes effectively to different spring-mesh systems for all phase-space coordinates.

notable for its high dimensionality and complex dynamical couplings, serves as a canonical testbed for learning physical systems, including deformable volumes and surfaces relevant to engineering applications (Pfaff et al., 2021). Node positions are updated via forces from spring extensions and compressions, resulting in nontrivial communication and computation patterns across the mesh. For a more in-depth description of the benchmark, refer to Appendix B.1. Unless stated otherwise, models are trained using teacher forcing for next-token prediction with a fixed context window of 30 time-steps. This choice aligns with prior configurations used in large-scale visual-language architectures (e.g., $\pi$0: 50 time-steps in Black et al. (2024), OpenVLA: 8 time-steps in Kim et al. (2025)). We also provide an ablation study on the context window, refer to Appendix D. Notably, our models are significantly smaller, yet demonstrate strong capability for long-term, temporally consistent predictions within this horizon as the results in Fig. 2 indicate.

## 4.2 Open Quantum System

To benchmark MetaSym on a novel open quantum dynamics task, we consider a parametric oscillator initialized in a coherent state (Gardiner & Zoller, 2010). The Hamiltonian includes a harmonic term $\hat{H}_{osc} = \omega \hat{a}^\dagger \hat{a}$, a squeezing term $\hat{H}_{sqz} = \frac{i\chi}{2}(\hat{a}^{\dagger 2} - \hat{a}^2)$, and a cubic drive $\hat{H}_{cubic} = \beta(\hat{a}^3 + \hat{a}^{\dagger 3})$, with $\hat{a}, \hat{a}^\dagger$ the ladder operators in a truncated Fock space of dimension $N$. Dissipation arises via coupling to a thermal bath with rate $\gamma$ and mean occupation $\tilde{n}_{th}$. Continuous heterodyne detection yields a stochastic master equation for the conditional state (Appendix B.2). The only experimentally accessible data are the real and imaginary quadratures $X$ and $P$ of the heterodyne current, extracted from single-shot trajectories, as the quantum state itself remains inaccessible. In this context, quadratures ($X$ and $P$) denote the non-commuting observables, serving as the quantum optical analogues to classical position and momentum (Milburn, 1987). We simulate this setup by numerically solving the stochastic master equation (Johansson et al., 2012), varying measurement efficiency $\eta$, squeezing strength $\chi$, cubic drive $\beta$, oscillator frequency $\omega$, bath occupation $\tilde{n}_{th}$, and initial states. Training and out-of-distribution inference details are in Appendix B.2. As shown in Fig. 3, MetaSym performs well in this strongly stochastic, non-classical regime.

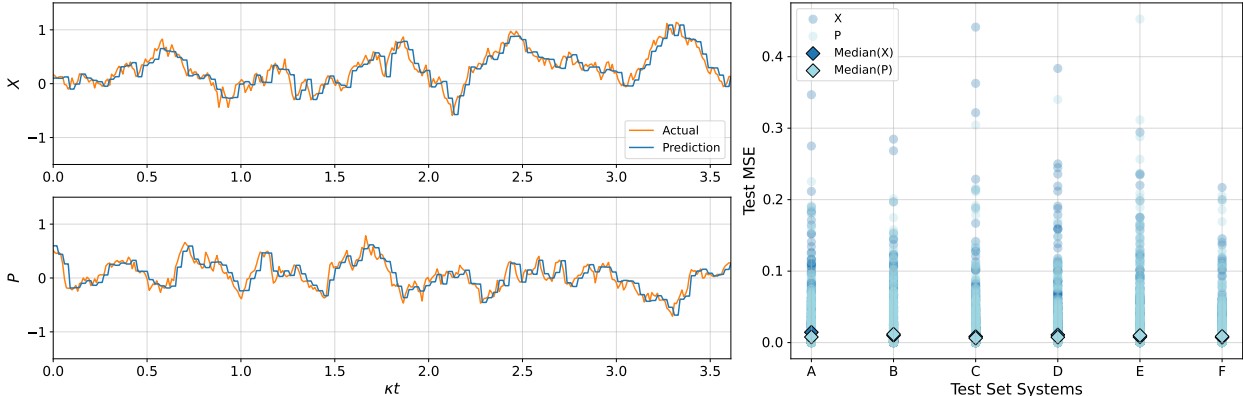

Figure 3: (**Top**) Time evolution of the two quadratures measured via heterodyne detection for a representative quantum system in the test set, characterized by a measurement efficiency ($\eta = 0.86$) and measurement strength ($\kappa$). The orange lines indicate the true measurement trajectories, while the blue lines display the model's predictions (MetaSym). The close overlap between these trajectories highlights the model's effectiveness in capturing the underlying quantum dynamics from heterodyne signals. The context window of the model is 30 time-steps. (**Bottom**) Plots showing the mean squared error (MSE) of **each of the predicted quadratures** (dots) for five randomly selected test systems. The consistently low median errors (boxes) across all tested systems underscore the robustness and generalization capabilities of the model.

Table 1: Trajectory error and parameter count against SOTA baselines for three OOD datasets

| Models | Spring-mesh System | | Quantum System | | Quadrotor | |
|---|---|---|---|---|---|---|
| | Param. Count | Traj. MSE ($\pm\sigma$) | Param. Count | Traj. MSE ($\pm\sigma$) | Param. Count | Traj. MSE ($\pm\sigma$) |
| DHNNs | 3.0M | 32.468 (28.086) | N/A | N/A | 3138 | 26.375 (9.160) |
| Transformer | 3.2M | 36.653 (15.618) | 194 | 0.950 (0.239) | 4680 | 34.584 (14.063) |
| Naive Baseline (MLP) | 3.5M | 323.199 (13.959) | 262 | 0.898 (0.226) | 4012 | 39.311 (13.937) |
| QLSTM (Flurin et al., 2020) | N/A | N/A | 244 | 0.891 (0.233) | N/A | N/A |
| Dissipative SymODEN | 3.2M | 32.054 (20.337) | N/A | N/A | 3187 | 29.174 (9.547) |
| FNO | 3.3M | 25.187 (24.223) | 33602 | 7.648 (0.411) | 3336 | 27.932 (11.157) |
| iMODE | 3.2M | 23.858 (11.552) | 202 | 7.667 (0.436) | 3204 | 57.190 (20.903) |
| **MetaSym (ours)** | **2.9M** | **19.233 (15.673)** | **130** | **0.859 (0.215)** | **3036** | **25.889 (8.967)** |

## 4.3 Quadrotor

For our final benchmark, we evaluate on a quadrotor, a canonical yet challenging system due to its floating-base dynamics and discontinuous orientation representations (Sanyal & Roy, 2023; Zhou et al., 2019). The floating base induces internal covariate shifts during training, as phase-space magnitudes could vary across time. Regarding its orientation representation, Euler angles suffer from singularities, while quaternions impose a unit-norm constraint that is typically handled via loss penalties, often leading to local minima and unstable training. To address this, we represent the pose of the quadrotor (i.e., position and orientation) using the tangent space of the $\mathbb{SE}(3)$ group (Solà et al., 2021), which avoids the explicit enforcement of the unit norm constraint. MetaSym is pretrained using simulation data for a variety of different systems described in detail in in Appendix B.3. Importantly, the pretrained network is then fine-tuned via the query-value weight adaptation technique introduced with the *ActiveDecoder* to a real-world dataset (Mohajerin & Waslander, 2018). The results not only demonstrate our method's efficacy and robustness on real-world systems but also the fast-adaptation capability of MetaSym that effectively diminishes the sim-to-real gap. Even though a detailed real-world deployment analysis is beyond the scope of the current paper, we performed an additional study to verify our model's inference time. The results indicate that MetaSym can run on an NVIDIA RTX4090 in **1.551 ms** for its full **30 time-step context-window**.

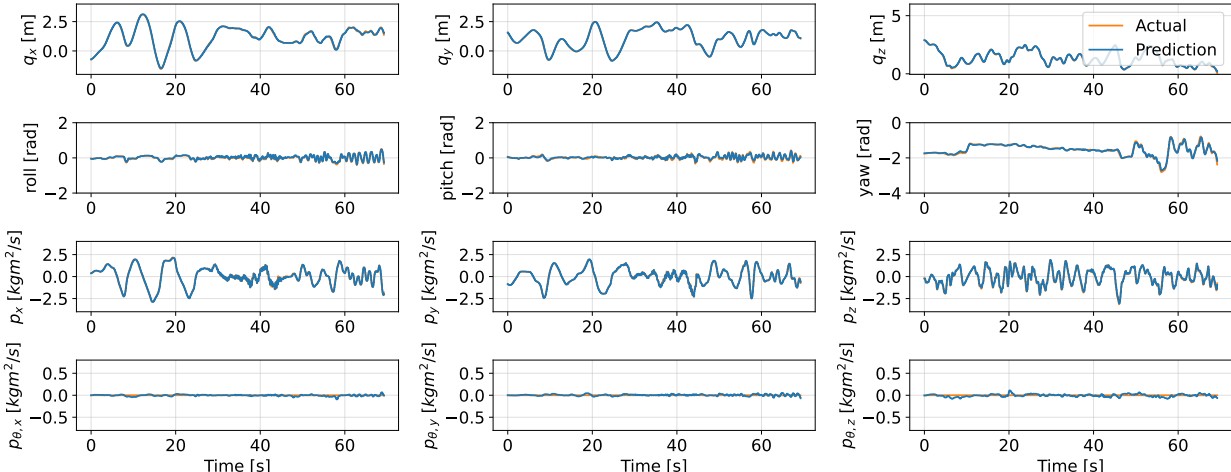

Figure 4: Represents the translational and angular phase-space evolution of the quadrotor, with training generated using the *Crocoddyl* trajectory optimization package (Mastalli et al., 2020). Each training system is initialized with randomized initial conditions and a randomized terminal position over a $1.5s$ horizon (150 time-steps) with a context window of 30 time-steps. The displayed test trajectories represent the evolution of a **real-world quadrotor**. The ground-truth test trajectory (orange line) overlaps with MetaSym's predictions (blue line) indicating the excellent predictive capabilities of our architecture. The MSE across 54 real-world test trajectories is $\mathbf{0.028 \pm 0.091}$ for a context-window of 30 time-steps. Note we translate the quaternions to roll-pitch-yaw angles to have better interpretability of the results.

### 4.4 Benchmarks

We benchmarked MetaSym against three baselines, an autoregressive Transformer (Vaswani, 2017), a physics-informed dissipative Hamiltonian neural network (DHNN) (Greydanus et al., 2019), a naive Multi-Layer Perceptron (MLP), the Dissipative SymODEN (Zhong et al., 2020), the Fourier neural operator (FNO) (Li et al., 2021), and iMODE (Li et al., 2023). Both MetaSym and the Transformer perform autoregressive next time-step prediction, whereas the DHNN models the system's symplectic and dissipative gradients, relying on an integrator for trajectory unrolling. The MLP also predicts the next state directly. Unlike the others, the DHNN cannot predict quadrature measurements due to ill-defined derivatives involving complex Wiener terms (Milburn, 1987), and its training proved less stable than that of MetaSym and the Transformer. All models were evaluated in an autoregressive roll-out regime. As shown in Table 1, MetaSym consistently outperforms the alternatives in long-horizon prediction accuracy, particularly when using a 30 time-step context window. It achieves lower error accumulation and requires fewer parameters, which is especially advantageous for high-dimensional systems like spring-meshes. The variance present across all models is attributed to the noise that we artificially added to each network's input in order to highlight MetaSym's robustness and real-world applicability. In quantum dynamics, where real-time prediction and control are constrained by hardware latency and memory (Reuer et al., 2023), MetaSym's compact architecture offers a significant advantage over Transformer-based architectures (Vaidhyanathan et al., 2024).

## 5    Conclusion

In this work, we introduced **MetaSym**, a novel deep-learning framework that combines structure-preserving symplectic networks with an autoregressive, decoder equipped with meta-learning for modeling a wide-range of physical systems. The core idea rests on striking a balance between *strong physical priors*, namely the intrinsic symplectic structure of Hamiltonian mechanics, and the *flexibility* required to capture non-conservative effects and heterogeneous system parameters. Our experimental results across diverse domains,

ranging from high-dimensional spring-mesh dynamics to open-quantum systems and **real-world** quadrotor systems, demonstrated that MetaSym outperforms SOTA baselines in both long-horizon accuracy and few-shot adaptation with smaller model sizes even for real-world systems.

The *SymplecticEncoder* approximates canonical flows while preserving key invariants, significantly mitigating energy drift and ensuring robust long-term predictions. The encoder's invertible design enforces time-reversal symmetry and reduces error accumulation. Meanwhile, the *ActiveDecoder* models departs from ideal Hamiltonian evolution through autoregressive prediction and meta-attention. The resulting architecture is computationally efficient, given that it does not require explicit numerical integration during inference, and through meta-learning, it readily adapts to system variations with minimal additional data. This approach offers a scalable and unified framework for high-fidelity physics modeling in realistic settings with provable near-symplectic properties (see Appendix C.4).

Additionally, while a substantial line of work has explored Meta-PINNs and parameter-aware operator learning to generalize across varying physical coefficients exist (Penwarden et al., 2023; Wang et al., 2021). These methods effectively adapt to new PDE instances, they typically enforce physical consistency via "soft" regularization terms in the loss function. This reliance on soft constraints often necessitates careful hyperparameter tuning of penalty weights and does not strictly guarantee the preservation of conservation laws during inference. In contrast, MetaSym enforces symplectic symmetries through "hard" architectural constraints within the encoder. This design ensures that the core conservative dynamics remain on the symplectic manifold by construction, independent of optimization stability, while relegating only the non-conservative dynamics to the adaptable decoder.

**Limitations & Future Work**. It is important to note that MetaSym's strongest guarantees (e.g. bounded energy drift, near-symplectic behavior, and long-horizon stability) rely on the availability of at least some representative conservative data for pretraining the SymplecticEncoder. When such data are unavailable, MetaSym can be interpreted as a hybrid physics-informed / data-driven model whose guarantees degrade continuously toward those of a fully black-box architecture. Considering MetaSym's promising performance, we seek to investigate several future directions such as incorporating a fully symplectic network for modeling realistic physics by exploiting the underlying structure of dissipation and our control signals. MetaSym's effectiveness on real-world system data also remains to be investigated. Another natural extension of few-shot adaptation is online learning and control, since MetaSym can quickly adapt to new system configurations with minimal data. This could be leveraged in real-time control loops and model-based Reinforcement Learning (RL) algorithms. Finally, future research could focus on how to integrate the decoder's adaptation step with RL or adaptive Model Predictive Control (MPC) frameworks, effectively enabling self-tuning controllers in rapidly-changing environments.

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

# A  Meta Learning Setup

As mentioned in Section 3, we elaborate on the algorithmic pipeline of our *SymplecticEncoder* and *ActiveDecoder* modules. In particular, we outline the MAML-based meta-learning strategy for the *SymplecticEncoder* and the bi-level adaptation meta-learning for the *ActiveDecoder*. Fig. 8 provides an interpretable insight into the effect of inner adaptation during the meta-update step and its effectiveness.

## A.1  Encoder

Following Section 3.1, Algorithm 1 provides a high-level overview of how the *SymplecticEncoder* is trained via MAML meta-learning. More specifically, the Model-Agnostic Meta-Learning (MAML) is a meta-learning framework designed to train models that can adapt quickly to new tasks using only a small amount of data. The core idea is to learn an set of parameters that is not task-specific, but task agnostic. As a model-agnostic method, it can be applied to any model trained with gradient descent, including neural networks for classification, regression (e.g., our case with MetaSym), reinforcement learning, and beyond. During meta-training, MAML simulates adaptation by sampling systems, performing inner-loop updates on each system, and then optimizing the initial parameters via meta-gradients computed across systems. This results in a model that learns how to learn, making it highly effective in few-shot scenarios where fast generalization from limited supervision is required.

The task-specific gradient steps are performed on parameters that are initialized from a shared-parameter set, denoted as $\theta_{SE}$, for each system. These inner-loop (fast adaptation) updates are applied to independent copies of the initial parameters, denoted $\theta_{SE}^{(i)}$, one for each task (i.e., each system $i$). Crucially, the original $\theta_{SE}$ must remain unchanged during these task-specific updates, as it serves as the point from which all adaptations are made.

Only after the fast adaptation steps across tasks are completed do we compute the meta-gradient, based on the post-adaptation performance on each task, and use it to update the original initialization $\theta_{SE}$ via the outer-loop (meta) update. This structure ensures that $\theta_{SE}$ is optimized for adaptability, enabling efficient transfer to unseen systems.

---

**Algorithm 1** Meta-Learning for the *SymplecticEncoder*

---

**Require:** $\mathcal{D} = \{\mathcal{D}^{(1)}, \ldots, \mathcal{D}^{(N)}\}$: Training data from $N$ related systems. Each $\mathcal{D}^{(i)}$ is a trajectory $\left(\mathbf{q}_t^{(i)}, \mathbf{p}_t^{(i)}, \mathbf{u}_t^{(i)}\right)_{t=1}^{T}$.

1: **for** epoch = 1 **to** $N_{\text{epochs}}$ **do**
2:     **for** each mini-batch of systems $B \subseteq \{1, \ldots, N\}$ **do**
3:         `optimizer_theta.zero_grad()`
4:         `inner_optimizer.zero_grad()`
5:         **for** each system $i \in B$ **do**
6:             **Split the trajectory $\mathcal{D}^{(i)}$ into:**
7:             $\mathcal{I}_{\text{adapt}} \subset \{1, \ldots, \mathcal{T}_{\text{adapt}}^{(i)}\}, \quad \mathcal{I}_{\text{meta}} = \{1, \ldots, \mathcal{T}_{\text{meta}}^{(i)}\} \setminus \mathcal{I}_{\text{adapt}}.$
8:             $\mathcal{L}_{\text{meta}} \leftarrow 0$
9:             **Fast Adaptation: Adapt system-specific parameters**
10:            $\theta_{SE}^{(i)} \leftarrow \theta_{SE}$`.clone().detach()`          ▷ Detach
11:            **for** $k = 1$ **to** $K$ **do**
12:              `inner_optimizer.zero_grad()`
13:              $\mathcal{L}_{\text{inner}}^{(i)} \leftarrow 0$
14:              **for** $t \in \mathcal{I}_{\text{adapt}}$ **do**
15:                 $\left(\hat{\mathbf{q}}_{t+1}^{(i)}, \hat{\mathbf{p}}_{t+1}^{(i)}\right) \leftarrow \Phi_{\theta_{\text{SE}}}\left(\mathbf{q}_t^{(i)}, \mathbf{p}_t^{(i)}, \mathbf{u}_t^{(i)}; \theta_{SE}^{(i)}\right)$
16:                 $\mathcal{L}_{\text{inner}}^{(i)} \leftarrow \mathcal{L}_{\text{inner}}^{(i)} + \left\| [\hat{\mathbf{q}}_{t+1}^{(i)}, \hat{\mathbf{p}}_{t+1}^{(i)}] - [\mathbf{q}_{t+1}^{(i)}, \mathbf{p}_{t+1}^{(i)}] \right\|^2$
17:              **end for**
18:              $(\mathcal{L}_{\text{inner}}^{(i)}/\mathcal{T}_{\text{adapt}}^{(i)})$`.backward()`
19:              `inner_optimizer.step()`          ▷ Update $\theta_{SE}^{(i)}$ only
20:            **end for**
21:            **Meta Update: Update the global parameters $\theta_{SE}$**
22:            **for** $t \in \mathcal{I}_{\text{meta}}$ **do**
23:              $\theta_{SE} \leftarrow \theta_{SE}^{(i)*}$`.clone()`          ▷ No detach
24:              $\left(\hat{\mathbf{q}}_{t+1}^{(i)}, \hat{\mathbf{p}}_{t+1}^{(i)}\right) \leftarrow \Phi_{\theta_{\text{SE}}}\left(\mathbf{q}_t^{(i)}, \mathbf{p}_t^{(i)}, \mathbf{u}_t^{(i)}; \theta_{SE}\right)$
25:              $\mathcal{L}_{\text{meta}} \leftarrow \mathcal{L}_{\text{meta}} + \left\| [\hat{\mathbf{q}}_{t+1}^{(i)}, \hat{\mathbf{p}}_{t+1}^{(i)}] - [\mathbf{q}_{t+1}^{(i)}, \mathbf{p}_{t+1}^{(i)}] \right\|^2$
26:            **end for**
27:         **end for**
28:         $(\mathcal{L}_{\text{meta}}/\mathcal{T}_{\text{meta}}^{(i)})$`.backward()`          ▷ Eq. 4
29:         `optimizer_theta.step()`          ▷ Update $\theta_{SE}$ only
30:     **end for**
31: **end for**

---

## A.2 Decoder

Algorithm 2 provides a high-level overview of how the *ActiveDecoder* is trained via meta-learning.

---

**Algorithm 2** Meta-Learning for the *ActiveDecoder*

---

**Require:** $\mathcal{D} = \{\mathcal{D}^{(1)}, \ldots, \mathcal{D}^{(N)}\}$: Training data from $N$ related systems. Each $\mathcal{D}^{(i)}$ is a trajectory of states $\left(\mathbf{q}_t^{(i)}, \mathbf{p}_t^{(i)}, \mathbf{u}_t^{(i)}\right)_{t=1}^{T}$.

1: **for** epoch = 1 **to** $N_{\mathrm{epochs}}$ **do**
2:     **for** each mini-batch of systems $B \subseteq \{1, \ldots, N\}$ **do**
3:         `optimizer_theta.zero_grad()`
4:         **for** each system $i \in B$ **do**
5:             **Split the trajectory $\mathcal{D}^{(i)}$ into:**
6:             $\mathcal{I}_{\mathrm{adapt}} \subset \{1, \ldots, \mathcal{T}_{\mathrm{adapt}}^{(i)}\}, \quad \mathcal{I}_{\mathrm{meta}} = \{1, \ldots, \mathcal{T}_{\mathrm{meta}}^{(i)}\} \setminus \mathcal{I}_{\mathrm{adapt}}.$
7:             **Inner loop: Adapt local parameters $\zeta_i$**
8:             **for** $k = 1$ **to** $K$ **do**
9:                 **for** $t \in \mathcal{I}_{\mathrm{adapt}}$ **do**
10:                   `inner_optimizer.zero_grad()`
11:                   $\zeta_i$`.randomize()`             ▷ Reinitialize $\zeta_i$
12:                   $\left(\hat{\mathbf{q}}_{t+1}^{(i)}, \hat{\mathbf{p}}_{t+1}^{(i)}\right) \leftarrow \Phi_{\theta_{\mathrm{AD}}}\left(\mathbf{q}_{t:c}^{(i)}, \mathbf{p}_{t:c}^{(i)}, \mathbf{u}_{t:c}^{(i)}; \theta_{AD}, \zeta_i\right)$
13:                   $\mathcal{L}_{\mathrm{inner}}^{(i)} \leftarrow \left\| [\hat{\mathbf{q}}_{t+1}^{(i)}, \hat{\mathbf{p}}_{t+1}^{(i)}] - [\mathbf{q}_{t+1}^{(i)}, \mathbf{p}_{t+1}^{(i)}] \right\|^2$
14:                   $\left(\mathcal{L}_{\mathrm{inner}}^{(i)}/\mathcal{T}_{\mathrm{adapt}}^{(i)}\right)$`.backward()`       ▷ Eq. 3.2
15:                   `inner_optimizer.step()`       ▷ Update $\zeta_i$ only
16:                 **end for**
17:             **end for**
18:             **Outer loop: Meta-update for the global parameters $\theta_{AD}$**
19:             `optimizer_theta.zero_grad()`       ▷ No update to $\zeta_i$ now
20:             $\mathcal{L}_{\mathrm{outer}}^{(i)} \leftarrow 0$
21:             **for** $t \in \mathcal{I}_{\mathrm{meta}}$ **do**
22:                 $\left(\hat{\mathbf{q}}_{t+1}^{(i)}, \hat{\mathbf{p}}_{t+1}^{(i)}\right) \leftarrow \Phi_{\theta_{\mathrm{AD}}}\left(\mathbf{q}_{t:c}^{(i)}, \mathbf{p}_{t:c}^{(i)}, \mathbf{u}_{t:c}^{(i)}; \theta_{AD}, \zeta_i^*\right)$
23:                 $\mathcal{L}_{\mathrm{outer}}^{(i)} \leftarrow \mathcal{L}_{\mathrm{outer}}^{(i)} + \left\| [\hat{\mathbf{q}}_{t+1}^{(i)}, \hat{\mathbf{p}}_{t+1}^{(i)}] - [\mathbf{q}_{t+1}^{(i)}, \mathbf{p}_{t+1}^{(i)}] \right\|^2$
24:             **end for**
25:             $\left(\mathcal{L}_{\mathrm{outer}}^{(i)}/\mathcal{T}_{\mathrm{meta}}^{(i)}\right)$`.backward()`       ▷ Eq. 3.2
26:             `optimizer_theta.step()`       ▷ Update $\theta_{AD}$ only
27:         **end for**
28:     **end for**
29: **end for**

---

# B   Experimental Setup

## B.1   Spring-mesh System

Spring networks are a simple proxy for numerous physical scenarios involving deformable solids and cloth (in computer graphics, mechanics, or robotics). Each pair of connected particles exchanges spring forces that depend on displacements from rest lengths. Viscous-damping terms further shape the evolution. While the individual dynamics (Hooke's law) are straightforward, the combination of hundreds of coupled springs can give rise to complex large-scale deformations and oscillations. By benchmarking on a spring-mesh, we can examine how MetaSym learns large deformations, wave propagation through a membrane, or the impact of damping.

The training dataset consists of 25 distinct spring-mesh systems, each characterized by a unique set of physical parameters, including spring stiffness, damping coefficients, and initial conditions as indicated by Table 2a. These parameters are sampled from a predefined distribution to ensure sufficient diversity within the training set. Each system is simulated over a time span of 2000 irregular time-steps, capturing the full trajectory of node displacements and momenta. The resulting dataset provides a rich representation of dynamical behaviors within the parameter space.

To assess generalization and robustness, we construct a test dataset comprising ten additional spring-mesh systems. Unlike the training set, the parameters for these systems are drawn from distributions that differ from those used during training as indicated by Table 2b, introducing a domain shift that mimics real-world variations. This OOD test set enables a rigorous evaluation of the model's ability to extrapolate beyond the observed training dynamics and adapt to unseen conditions. For further information regarding the spring-mesh benchmark refer to Otness et al. (2021).

By incorporating both in-distribution training data and OOD validation data, this experimental setup ensures a comprehensive assessment of the model's learning capacity, robustness, and generalization performance when applied to novel physical configurations.

Table 2: Parameter ranges for in-distribution and out-of-distribution regimes.

(a) In-distribution parameters $T = 2000$

| Parameters | Values |
|---|---|
| $\gamma_{decay}$ [kg/s] | Uniform$(0.1, 0.2)$ |
| mass [kg] | Uniform$(0.1, 2.0)$ |
| $K_{spring}$ [N/m] | Uniform$(0.001, 0.5)$ |
| Init. Conds. [m] | Uniform$(0, 0.6)$ |
| dt [s] | Uniform$(0.001, 0.03)$ |
| Mesh Size $[n_x \times n_y]$ | $10 \times 10$ |

(b) Out-of-distribution parameters $T = 2000$

| Parameters | Values |
|---|---|
| $\gamma_{decay}$ [kg/s] | Uniform$(0.01, 0.05)$ |
| mass [kg] | Uniform$(3.0, 5.0)$ |
| $K_{spring}$ [N/m] | Uniform$(1.0, 3.0)$ |
| Init. Conds. [m] | Uniform$(0.9, 2.5)$ |
| dt [s] | Uniform$(0.1, 0.3)$ |
| Mesh Size $[n_x \times n_y]$ | $10 \times 10$ |

## B.2 Open Quantum System Derivation

Understanding the behavior of quantum systems plays a vital role in the development of promising technologies such as quantum computing, metrology, and sensing. While deep-learning has found great success in several areas such as quantum control (Vaidhyanathan et al., 2024), error correction (Bausch et al., 2024) and tuning (Ares, 2021; Gebhart et al., 2023), predicting the measurement record based on modeling quantum dynamics has long remained elusive. In many scenarios, the system of interest is *open*: it couples to an environment (or bath) that can introduce thermal noise and dissipation. Furthermore, continuous monitoring (e.g., via homodyne detection) adds additional *measurement backaction*, reflecting fundamental constraints from quantum mechanics (Jacobs & Steck, 2006). Capturing these noise and measurement effects is pivotal for accurately predicting quantum trajectories and devising robust control protocols.

Unlike closed Hamiltonian evolutions, open quantum systems require one to solve Stochastic Master Equations (SMEs) incorporating decoherence and measurement terms. These equations produce trajectories of the (mixed) quantum state conditioned on the noisy measurement record. In many practical settings, however, we only have direct access to certain observables (e.g., position and momentum quadratures) rather than the full quantum state. Hence, training a deep learning network to model the quantum system and to predict future measurement outcomes becomes a natural and practically relevant challenge. The SME describes the evolution of the *conditioned* quantum state $\rho_c(t)$ under the effect of environmental and measurement noise as in Jacobs & Steck (2006)

$$d\rho_c(t) = -i[H, \rho_c(t)]dt + \sum_j \mathcal{D}[\hat{L}_j]\rho_c(t)dt + \sqrt{\eta}\mathcal{H}[\hat{M}]\rho_c(t)dW_t, \tag{7}$$

where the *dissipator* is $\mathcal{D}[\hat{L}]\rho = \hat{L}\rho\hat{L}^\dagger - \frac{1}{2}\{\hat{L}^\dagger\hat{L}, \rho\}$. Each Linblad operator, $\hat{L}_j$, represents a *collapse operator* that encodes coupling to the environment (e.g., photon loss to a reservoir, thermal excitations, dephasing, etc.) (Manzano, 2020). The stochastic backaction term, $\mathcal{H}[\hat{M}]\rho = \hat{M}\rho + \rho\hat{M}^\dagger - \text{Tr}\left[\left(\hat{M} + \hat{M}^\dagger\right)\rho\right]\rho$, describes continuous monitoring of an observable $\hat{M}$, with $\eta$ the measurement efficiency ($0 \leq \eta \leq 1$). The Wiener increment, $dW_t$, captures the randomness inherent in quantum measurement outcomes.

### B.2.1 Setup for Parameteric Oscillator

In Section 4.2, we focus on a single-mode bosonic system with annihilation operator $\hat{a}$ and creation operator $\hat{a}^\dagger$. The Hamiltonian is:

$$H = \omega\hat{a}^\dagger\hat{a} + \frac{i\chi}{2}\left(\hat{a}^{\dagger 2} - \hat{a}^2\right) + \beta\left(\hat{a}^3 + \hat{a}^{\dagger 3}\right), \tag{8}$$

with $\omega$, $\chi$ and $\beta$ being the oscillator frequency, squeezing strength and cubic driving term respectively. We also include two Linblad operators,

$$\hat{L}_1 = \sqrt{\gamma\left(\bar{n}_{th} + 1\right)}\hat{a}, \quad \hat{L}_2 = \sqrt{\gamma\bar{n}_{\text{th}}}\hat{a}^\dagger, \tag{9}$$

where $\gamma$ is the coupling rate to the thermal bath, and $\bar{n}_{\text{th}}$ is the average occupation number.

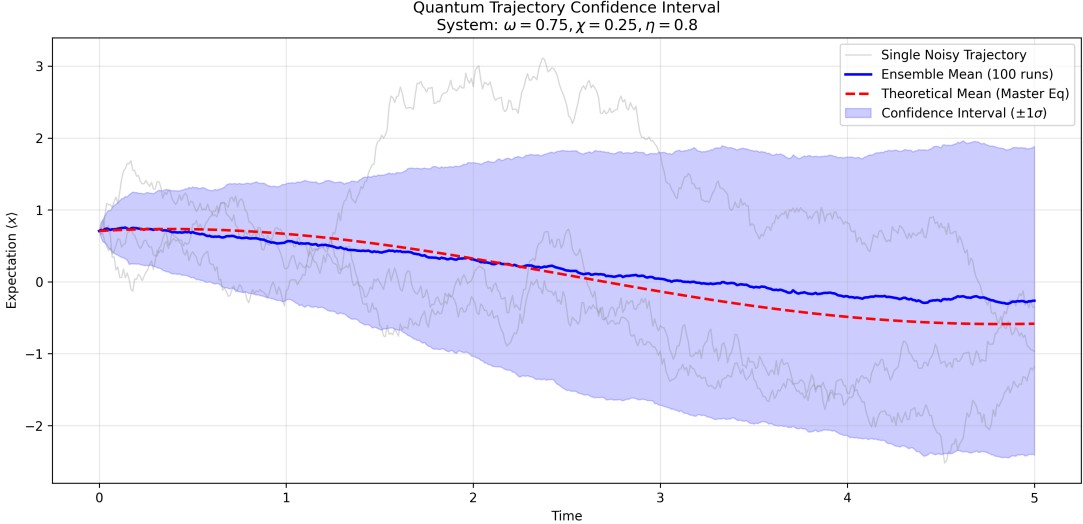

Figure 5: Quantum System's position quadrature $X$ evolution: The faint gray traces represent individual single-shot trajectories, showing the inherent randomness caused by measurement backaction and thermal noise, directly visualizing the noisy nature of the dataset. The blue shaded region marks the confidence interval, quantifying the standard deviation of the quantum state's fluctuations over time. This demonstrates the jagged unpredictability of individual runs. The ensemble average (solid blue line) converges to the smooth, and deterministic prediction of the Master Equation (red dashed line) over multiple trajectories.

A common measurement technique often employed in experimental settings is called heterodyne measurement. Heterodyne detection continuously measures both quadratures of the output of our dissipative system by mixing it with a local oscillator at a slightly shifted frequency and then demodulating the resulting beat signal. This yields two simultaneous photocurrents, often referred to as the in-phase ($I$) and quadrature ($Q$) components. This reflects a key practical outcome of solving Eq. 7 is that $\rho_c(t)$ depends on this random measurement trajectory.

We employ heterodyne detection of the field operator $\hat{a}$. Based on this measurement scheme, we can get separate the real and imaginary parts to obtain quadrature values that roughly correspond to $X$ and $P$ while adding quantum noise and measurement uncertainties due to quantum mechanical effect (Flurin et al., 2020).

In the following Tables 3b & 5a, we describe the parameters used to generate our dataset by solving the SME in order to generate training data.

Table 3: Parameter ranges used in training and evaluation of quantum trajectories. The in-distribution set represents conditions seen during training, while the out-of-distribution set introduces significant shifts in frequency, squeezing, thermal noise, and measurement efficiency.

(a) In-distribution parameters. Time step $dt = 0.5$, total duration $T = 600$.

| Parameter | Value |
|---|---|
| oscillator frequency $\omega$ | $\text{Uniform}(0.5, 1.0)$ |
| squeezing strength $\chi$ | $\text{Uniform}(0.1, 0.4)$ |
| thermal occup. $\langle n_{th} \rangle$ | $\text{Uniform}(0.1, 0.5)$ |
| meas. efficiency $\eta$ | $\text{Uniform}(0.7, 1.0)$ |

(b) Out-of-distribution parameters. Time step $dt = 0.5$, total duration $T = 600$.

| Parameter | Value |
|---|---|
| oscillator frequency $\omega$ | $\text{Uniform}(0.1, 0.4)$ |
| squeezing strength $\chi$ | $\text{Uniform}(0.5, 0.8)$ |
| thermal occup. $\langle n_{th} \rangle$ | $\text{Uniform}(0.6, 0.7)$ |
| meas. efficiency $\eta$ | $\text{Uniform}(0.4, 0.6)$ |

We also include, Fig. 5, that illustrates the stochastic evolution of the position quadrature $X$ for our system. The nature of the system's stochasticity is revealed by the shaded confidence interval, the shape of which is dictated by Eq. 7.

### B.3 Quadrotor

The quadrotor system challenges data-driven methods with its floating-base dynamics. To generate the training and OOD validation datasets we use *Crocoddyl* trajectory optimization package based on the dynamics model proposed by Geisert & Mansard (2016). The training dataset comprises 30 systems with randomized parameters such as inertia, torque constant and rotor lengths as indicated in Table 4a. Each trajectory is generated from a randomized initial condition to a random terminal position with zero velocity, within a pre-set bounding box, to avoid unrealistic velocities.

In the same manner the OOD validation set contains 10 trajectories, each one corresponding to a system with parameters drawn from the distributions indicated in Table 4b.

Table 4: Parameter distributions for simulating quadrotor dynamics. The in-distribution set covers training conditions, while the out-of-distribution set reflects variations in inertia, mass, center of gravity shift, and torque coupling beyond the training domain.

(a) In-distribution parameters. Time step $dt = 0.01$, duration $T = 250$.

| Parameter | Value |
|---|---|
| Inertia $\mathcal{I}_{\text{diag}}$ | $\text{Uniform}(0.1, 0.2)$ |
| mass $m$ | $\text{Uniform}(0.5, 3.0)$ |
| $d_{\text{cogs}}$ | $\text{Uniform}(0.2, 0.5)$ |
| $C_{\text{torques}}$ | $\text{Uniform}(0.001, 0.1)$ |

(b) Out-of-distribution parameters. Time step $dt = 0.01$, duration $T = 500$.

| Parameter | Value |
|---|---|
| Inertia $\mathcal{I}_{\text{diag}}$ | $\text{Uniform}(0.3, 0.7)$ |
| mass $m$ | $\text{Uniform}(3.0, 5.0)$ |
| $d_{\text{cogs}}$ | $\text{Uniform}(0.5, 0.7)$ |
| $C_{\text{torques}}$ | $\text{Uniform}(0.1, 0.2)$ |

The results in Fig. 6 verify the accuracy and effectiveness of our method for simulated floating-base dynamics systems. This consists the pre-training executed before fine-tuning the architecture on the real-world quadrotor dataset illustrated in the main text.

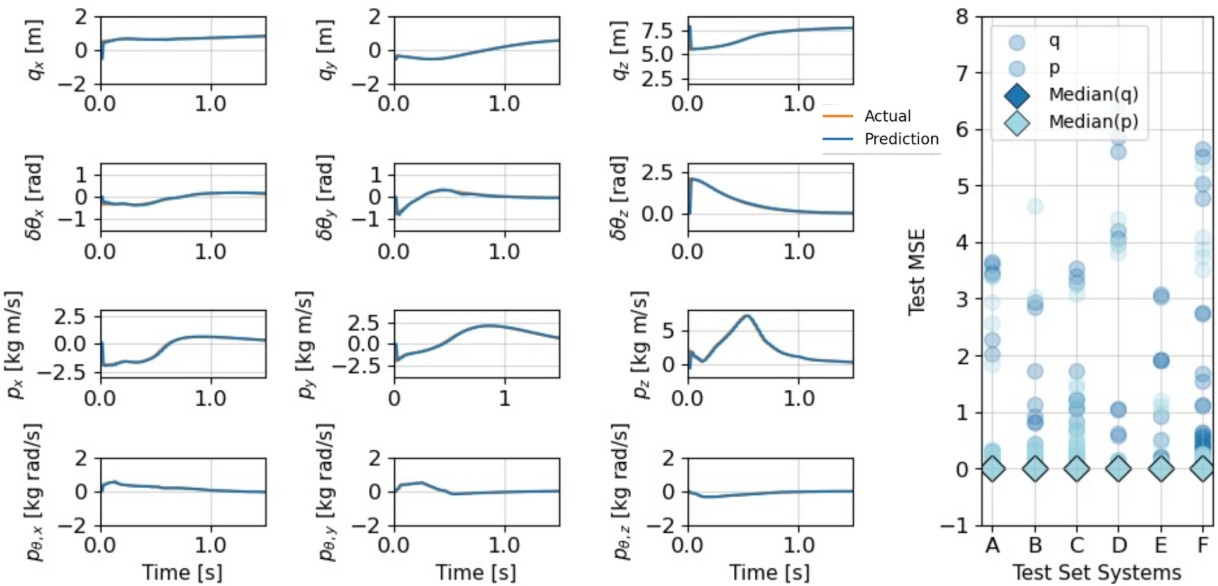

Figure 6: **(Left)** Represents the translational and angular phase-space evolution of the quadrotor, with training and test trajectories generated using the *Crocoddyl* trajectory optimization package (Mastalli et al., 2020). Each task is initialized with randomized initial conditions and a randomized terminal position over a $1.5s$ horizon (150 time-steps) with a context window of 30 time-steps. The ground-truth test trajectory (orange line) overlaps with MetaSym's predictions (blue line) indicating the excellent predictive capabilites of our model. **(Right)** Plots summarizing the mean squared error (MSE) of **each of the phase-space coordinates evolution** for five randomly generated test systems (dots). The consistently low median errors (boxes) across all components underscore the robustness and generalization capabilities of the model.

## C $O(\rho)$ **Near-Symplectic Architecture**

In this section, we prove that composing our *SymplecticEncoder* $\Phi_{\theta_{SE}}$ with our *ActiveDecoder* $\Phi_{\theta_{AD}}$ handling control signals and dissipative effects yields a near-symplectic map, with an explicit bound on preserving approximate symplectic geometry even when tested in adverse dissipative conditions. This formalizes the realistic effects we can introduce to MetaSym and the inductive bias provided by symplectic invariants.

### C.1 Overall Map

The full transformation is

$$(\mathbf{q}_t, \mathbf{p}_t) \ \mapsto \ \mathbf{z}_c = \Phi_{\theta_{SE}}(\mathbf{q}_t, \mathbf{p}_t) \ \mapsto \ \mathbf{z}_d = \Phi_{\theta_{AD}}(\mathbf{z}_c, \mathbf{u}_t, \mathbf{d}_t).$$

Hence, at the *global* level, we define

$$\Phi_\theta(\mathbf{q}_t, \mathbf{p}_t, \mathbf{u}_t, \mathbf{d}_t) := \Phi_{\theta_{AD}}\big(\Phi_{\theta_{SE}}(\mathbf{q}_t, \mathbf{p}_t), \ \mathbf{u}_t, \ \mathbf{d}_t\big).$$

We aim to show that if $\Phi_{\theta_{AD}}$ remains a *small* (bounded) perturbation $O(\rho)$ from identity in $\mathbf{z}_c$-space, then $\Phi_\theta$ preserves the symplectic structure up to a small error term. This is called $O(\rho)$ *near-symplecticity*. We assume that the dissipation $\mathbf{d}_t$ and control-input $\mathbf{u}_t$ are separable for the sake of this proof.

### C.2 SymplecticEncoder

From Jin et al. (2020), we know that *LA-SympNets* are fully symplectic due to their construction. By extension, we can show that our *SymplecticEncoder* $\Phi_{\theta_{SE}}$ is symplectic.

**Definition C.1** (Symplectic Property). Let $\mathcal{X} = \mathbb{R}^{2d}$ represent the canonical phase space with coordinates $\mathbf{x} \in (\mathbf{q}, \mathbf{p})$. We write

$$\mathbf{J} = \begin{pmatrix} \mathbf{0} & \mathbf{I}_d \\ -\mathbf{I}_d & \mathbf{0} \end{pmatrix},$$

the standard $2d \times 2d$ symplectic matrix. A differentiable map $\Psi : \mathcal{X} \to \mathcal{X}$ is *strictly symplectic* if

$$\mathrm{d}\Psi(\mathbf{x})^\top \mathbf{J} \, \mathrm{d}\Psi(\mathbf{x}) = \mathbf{J} \quad \forall \mathbf{x} \in \mathcal{X}.$$

This implies $\det(\mathrm{d}\Psi) = 1$ (volume preservation).

We know

$$\Phi_{\theta_{SE}} : \; \mathcal{X} \; \to \; \mathcal{Z} \subseteq \mathbb{R}^{2d}$$

is strictly symplectic, i.e.

$$\mathrm{d}\Phi_{\theta_{SE}}(\mathbf{x})^\top \mathbf{J} \, \mathrm{d}\Phi_{\theta_{SE}}(\mathbf{x}) = \mathbf{J}, \quad \det(\mathrm{d}\Phi_{\theta_{SE}}(\mathbf{x})) = 1.$$

Internally, since $\Phi_{\theta_{SE}}$ is a composition of symplectic sub-blocks (e.g. LA-SympNets). Its parameters, $\theta_{SE}$, can be partially meta-learned as long as the strict symplectic property is retained.

## C.3 Decoder with Control and Dissipation

We define

$$\Phi_{\theta_{AD}} : \quad \mathcal{Z} \times \mathcal{U} \times \mathcal{D} \; \to \; \mathcal{Z},$$

where:

- $\mathcal{Z} \subseteq \mathbb{R}^{2d}$ is the latent phase-space output by ($\Phi_{\theta_{SE}}$),

- $\mathcal{U} \subseteq \mathbb{R}^m$ represents *control signals* (bounded by $\|\mathbf{u}_t\| \leq U_{\max}$),

- $\mathcal{D} \subseteq \mathbb{R}^r$ represents *dissipative parameters* (bounded by $\|\mathbf{d}_t\| \leq D_{\max}$), which model forces that remove or drain energy (e.g., friction or drag).

In order to account for the effects of the cross attention, the decoder modifies the latent state by

$$\Phi_{\theta_{AD}}(\mathbf{z}_c, \mathbf{u}_t, \mathbf{d}_t) = \mathbf{z}_c + F_{\theta_{AD}}(\mathbf{z}_c, \mathbf{u}_t, \mathbf{d}_t),$$

where ($F_{\theta_{AD}}$) can be cross-attention with magnitude modulated by $\|\mathbf{u}_t\|$, or a damping formula modulated by $\|\mathbf{d}_t\|$.

## C.4 $O(\rho)$ Near-Symplectic Proof and Explicit Bound

In this section, we prove that if $\mathbf{z}_c \mapsto \mathbf{z}_c + F_{\theta_{AD}}(\mathbf{z}_c)$ is a *bounded perturbation* from identity (in partial derivatives), then the composition with a strict symplectic map remains close to preserving $\mathbf{J}$.

**Bounded Perturbation Assumption.** For the *ActiveDecoder* map, we take the partial derivative w.r.t. $\mathbf{z}_c$ in $F_{\theta_{AD}}$ is *bounded* by a linear-type function of ($\|\mathbf{u}_t\|, \|\mathbf{d}_t\|$):

$$\left\| \frac{\partial F_{\theta_{AD}}}{\partial \mathbf{z}_c}(\mathbf{z}_c, \mathbf{u}_t, \mathbf{d}_t) \right\| \leq \alpha_0 + \alpha_u \|\mathbf{u}_t\| + \alpha_d \|\mathbf{d}_t\|,$$

for some constants $\alpha_0, \alpha_u, \alpha_d \geq 0$. This covers *cross-attention* scaled by $\|\mathbf{u}_t\|$ and *dissipative* scaled by $\|\mathbf{d}_t\|$ terms. Since $\|\mathbf{u}_t\| \leq U_{\max}$ and $\|\mathbf{d}_t\| \leq D_{\max}$, we define:

$$\rho := \alpha_0 + \alpha_u U_{\max} + \alpha_d D_{\max}.$$

Hence, $\max_{\mathbf{z}_c, \mathbf{u}_t, \mathbf{d}_t} \left\| \mathrm{d}_{\mathbf{z}_c} \Phi_{\theta_{AD}} - \mathbf{I} \right\| \leq \rho$. Equivalently,

$$\left\| \mathrm{d}_{\mathbf{z}_c} F_{\theta_{AD}}(\mathbf{z}_c, \mathbf{u}_t, \mathbf{d}_t) \right\| \leq \rho, \quad \forall (\mathbf{z}_c, \mathbf{u}_t, \mathbf{d}_t).$$

**The Composed Map.**    Recall we define the global map

$$\Phi_\theta(\mathbf{q}_t, \mathbf{p}_t, \mathbf{u}_t, \mathbf{d}_t) \;=\; \Phi_{\theta_{AD}}\!\Big(\Phi_{\theta_{SE}}(\mathbf{q}_t, \mathbf{p}_t),\; \mathbf{u}_t,\; \mathbf{d}_t\Big).$$

Writing $\mathbf{x} = (\mathbf{q}_t, \mathbf{p}_t) \in \mathbb{R}^{2d}$ for convenience, we have

$$\mathbf{z}_c = \Phi_{\theta_{SE}}(\mathbf{x}) \quad \text{and} \quad \mathbf{z}_d = \Phi_{\theta_{AD}}(\mathbf{z}_c,\; \mathbf{u}_t,\; \mathbf{d}_t).$$

To show near-symplecticity, we study

$$\mathrm{d}\Phi_\theta(\mathbf{x}, \mathbf{u}_t, \mathbf{d}_t) = \underbrace{\mathrm{d}_{\mathbf{z}_c}\Phi_{\theta_{AD}}\big(\mathbf{z}_c, \mathbf{u}_t, \mathbf{d}_t\big)}_{\mathbf{I}+\mathbf{A},\; \|\mathbf{A}\|\leq\rho} \;\;\times\;\; \underbrace{\mathrm{d}\Phi_{\theta_{SE}}(\mathbf{x})}_{\text{strictly symplectic.}}$$

**Theorem C.1** ($O(\rho)$ Near-Symplectic Composition)**.**  Suppose $\Phi_{\theta_{SE}}$ is strictly symplectic, i.e.

$$\mathrm{d}\Phi_{\theta_{SE}}(\mathbf{x})^\top \, \mathbf{J} \, \mathrm{d}\Phi_{\theta_{SE}}(\mathbf{x}) = \mathbf{J} \quad \text{and} \quad \det(\mathrm{d}\Phi_{\theta_{SE}}(\mathbf{x})) = 1.$$

Also assume $\mathrm{d}_{\mathbf{z}_c}\Phi_{\theta_{AD}}$ satisfies the bounded-perturbation condition $\max \|\mathrm{d}_{\mathbf{z}_c}\Phi_{\theta_{AD}} - I\| \leq \rho$ over $\|\mathbf{u}_t\| \leq U_{\max}$, $\|\mathbf{d}_t\| \leq D_{\max}$. Then for the composed map $\Phi_\theta$, we have:

$$\big\|\mathrm{d}\Phi_\theta(\mathbf{x})^\top \, \mathbf{J} \, \mathrm{d}\Phi_\theta(\mathbf{x}) - \mathbf{J}\big\| \;\leq\; C\,\rho,$$

for a constant $C > 0$ depending on the norm of $\mathrm{d}\Phi_{\theta_{SE}}(\mathbf{x})$. Hence $\Phi_\theta$ is $\epsilon$-symplectic with $\epsilon = C\,\rho$. Furthermore,

$$\det\big(\mathrm{d}\Phi_\theta(\mathbf{x})\big) = 1 + O(\rho),$$

implying near-volume preservation as well.

*Proof.*  Let $\mathbf{x} = (\mathbf{q}_t, \mathbf{p}_t)$, $\mathbf{z}_c = \Phi_{\theta_{SE}}(\mathbf{x})$, and $\mathbf{I} + \mathbf{A} = \mathrm{d}_{\mathbf{z}_c}\Phi_{\theta_{AD}}(\mathbf{z}_c, \mathbf{u}_t, \mathbf{d}_t)$ with $\|A\| \leq \rho$. Then

$$\mathrm{d}\Phi_\theta(\mathbf{x}, \mathbf{u}_t, \mathbf{d}_t) = (\mathbf{I} + \mathbf{A}) \, \mathrm{d}\Phi_{\theta_{SE}}(\mathbf{x}).$$

Hence

$$\mathrm{d}\Phi_\theta^\top \, \mathbf{J} \, \mathrm{d}\Phi_\theta = \mathrm{d}\Phi_{\theta_{SE}}^\top \, (\mathbf{I} + \mathbf{A})^\top \, \mathbf{J} \, (\mathbf{I} + \mathbf{A}) \, \mathrm{d}\Phi_{\theta_{SE}}.$$

Expanding $(\mathbf{I} + \mathbf{A})^\top \mathbf{J} (\mathbf{I} + \mathbf{A}) = \mathbf{J} + \mathbf{A}^\top \mathbf{J} + \mathbf{J}\mathbf{A} + \mathbf{A}^\top \mathbf{J}\mathbf{A} = \mathbf{J} + O(\|\mathbf{A}\|)$, we substitute $\mathrm{d}\Phi_{\theta_{SE}}^\top \, \mathbf{J} \, \mathrm{d}\Phi_{\theta_{SE}} = \mathbf{J}$:

$$\mathrm{d}\Phi_\theta^\top \, \mathbf{J} \, \mathrm{d}\Phi_\theta = \mathbf{J} + O(\|A\|) = \mathbf{J} + O(\rho).$$

In operator norm, $\|\mathrm{d}\Phi_\theta^\top \, \mathbf{J} \, \mathrm{d}\Phi_\theta - \mathbf{J}\| \leq C\,\rho$. Since $\mathrm{d}\Phi_{\theta_{SE}}$ is volume-preserving, $\det(\mathrm{d}\Phi_\theta) = \det(\mathbf{I} + \mathbf{A}) \odot \mathbf{1} = \mathbf{1} + O(\rho)$. $\qquad\square$

**Explicit Cross-Attention Bound.**    For instance, if $F_{\theta_{AD}}$ includes a *cross-attention* term scaled by $\|\mathbf{u}_t\|$ plus a *dissipative* term scaled by $\|\mathbf{d}_t\|$, we might write

$$F_{\theta_{AD}}(\mathbf{z}_c, \mathbf{u}_t, \mathbf{d}_t) = \alpha\,\mathrm{CrossAttn}(\mathbf{z}_c, \mathbf{u}_t) \;-\; \gamma\,(\mathbf{d}_t^\top \mathbf{z}_c),$$

where $\mathbf{d}_t^\top \mathbf{z}_c$ indicates some parametric dissipator. If CrossAttn has partial derivative in $\mathbf{z}_c$ normed by $\|\mathbf{u}_t\|$, and $\mathbf{d}_t^\top \mathbf{z}_c$ is linear in $\|\mathbf{d}_t\|$, then

$$\left\| \frac{\partial F_{\theta_{AD}}}{\partial \mathbf{z}_c} \right\| \;\leq\; \alpha_0 \;+\; \alpha_u \|\mathbf{u}_t\| \;+\; \alpha_d \|\mathbf{d}_t\|,$$

giving the same $\rho = \alpha_0 + \alpha_u U_{\max} + \alpha_d D_{\max}$.

We choose not to explicitly bound this perturbation during training to allow for MetaSym to model extremely dissipative systems. In Appendix D.2, we provide the perturbation bound $C\rho$ empirically for systems undergoing extreme dissipation and control. We observe that this bound is $< 1$ even under these extreme settings.

# D   Ablation Studies

To rigorously assess the efficacy of our proposed framework relative to current SOTA methods, we conduct a series of ablation studies. All experiments are performed on a moderately complex yet tractable spring-mesh system consisting of a $3 \times 3$ node grid. Unless otherwise noted, all other hyperparameters remain identical to those in Tables 2a & 2b.

## D.1   Modeling Conservative Systems

The *SymplecticEncoder* is designed to explicitly preserve symplectic structure by learning a generalized latent, symplectic basis, which captures the conservative phase-space dynamics. By design, it conserves invariant quantities such as energy and volume in phase space (Jin et al., 2020). The *ActiveDecoder* models system-parameter variations and captures dissipative and control-related effects as bounded perturbations to the symplectic manifold (see Section C.4). Hence the architecture is equipped to model fully-conservative systems and adapt to changing parameters. We conduct two different experiments in order to verify the efficacy of our architecture and the validity of our claims.

### D.1.1   Conservative Spring-Mesh System

We test our architecture against a conservative spring-mesh at test time. Specifically, the *ActiveDecoder* has been pre-trained on dissipative data and we adapt it using our meta-learning procedure to 25 fully conservative spring meshes. The achieved trajectory MSE across these systems is $\mathbf{0.050 \pm 0.016}$. This demonstrates that our meta-learning paradigm can accommodate substantial variations in system parameters through the *ActiveDecoder*. Simultaneously, the architecture respects the relevant physical invariances.

### D.1.2   Conservative Harmonic Oscillator

A harmonic oscillator is the prototypical example taught in many undergraduate courses in dynamics. It constitutes a well-studied system with analytical solutions. Stability analysis and dynamics' research (Strogatz, 2015) has equipped us with the tools necessary to study these linear systems in depth. By extension, we can understand further our architecture's true predictive capabilities in terms of physical invariances, such as energy and phase-space volume conservation.

The Ordinary-Differential Equation (ODE) modelling the harmonic oscillator's dynamics in phase-space coordinates is considered in Eq. 10,

$$\frac{d}{dt}\begin{pmatrix} x(t) \\ p(t) \end{pmatrix} = \begin{pmatrix} 0 & \frac{1}{m} \\ -m\omega^2 & 0 \end{pmatrix}\begin{pmatrix} x(t) \\ p(t) \end{pmatrix}, \tag{10}$$

where $\omega = \sqrt{\frac{k}{m}}$ is the oscillator frequency with mass $m$ and spring constant $k$. The position is indicated by $x(t)$ and the momentum coordinate by $p(t)$. The analytical solution for this is given by Eq. 11,

$$\begin{pmatrix} x(t) \\ p(t) \end{pmatrix} = \begin{pmatrix} \cos(\omega t) & \frac{1}{m\omega}\sin(\omega t) \\ -m\omega\sin(\omega t) & \cos(\omega t) \end{pmatrix}\begin{pmatrix} x_0 \\ p_0 \end{pmatrix}, \tag{11}$$

where $(x_0, p_0)$ are the initial conditions.

We train MetaSym with harmonic oscillators with varying masses and spring constants depicted in Table 5b. The results in terms of phase-space trajectories, energy-conservation are highlighted in Fig. 7.

Beyond energy conservation, Hamiltonian dynamics preserve phase-space volume according to Liouville's theorem. For the one-dimensional harmonic oscillator, this implies that the area enclosed by any closed orbit in the $(x, p)$ phase-space must remain invariant over time. To quantitatively assess whether MetaSym respects this structural property, we compute the phase-space area enclosed by both the ground-truth and predicted trajectories using a discrete line-integral (shoelace) estimator. More specifically, the enclosed phase-space area of a closed discrete trajectory $\{(q_i, p_i)\}_{i=0}^{N-1}$ is computed as $A_N = \frac{1}{2}\left|\sum_{i=0}^{N-1}(q_i p_{i+1} - q_{i+1} p_i)\right|$. In

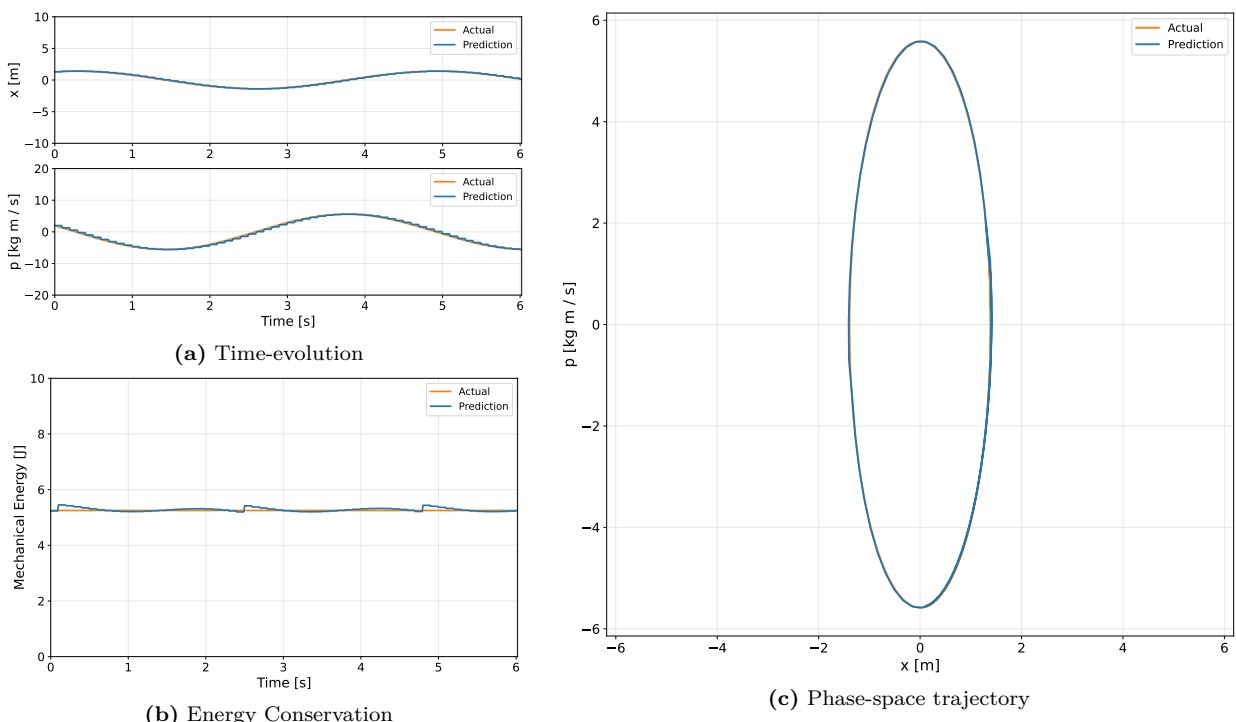

**(a)** Time-evolution

**(b)** Energy Conservation

**(c)** Phase-space trajectory

Figure 7: (a) MetaSym predicts the dynamics of the harmonic oscillator with high accuracy, since the prediction overlaps with the ground truth. (b) The energy drift of MetaSym during autoregressive rollouts remains negligible and uniformly bounded, as evidenced by the small, repetitive oscillations observed over time rather than any secular growth. (c) The overlapping prediction and ground-truth closed-orbits further support our claims of physical-invariance preservation.

the continuum limit $N \to \infty$, this converges to the symplectic line integral $A = \oint p\, dq = \iint dq \wedge dp$, corresponding to the symplectic area enclosed by the orbit.

Table 5: Training and out-of-distribution testing parameter ranges for the harmonic oscillator.

(a) In-distribution parameters. Time step $dt = 0.01$, total duration $T = 3000$.

| Parameter | Value |
|---|---|
| Mass $(m\ [kg])$ | Uniform$(0.5, 1.0)$ |
| Spring Const. $(k\ [N/m])$ | Uniform$(0.5, 4.0)$ |
| Init. Pos. $(x_0\ [m])$ | Uniform$(-1.0, 1.0)$ |
| Init. Vel. $(v_0\ [m/s])$ | Uniform$(-1.0, 1.0)$ |

(b) Out-of-distribution parameters. Time step $dt = 0.01$, total duration $T = 800$.

| Parameter | Value |
|---|---|
| Mass $(m\ [kg])$ | Uniform$(2.5, 3.0)$ |
| Spring Const. $(k\ [N/m])$ | Uniform$(5.0, 6.0)$ |
| Init. Pos. $(x_0\ [m])$ | Uniform$(1.0, 1.5)$ |
| Init. Vel. $(v_0\ [m/s])$ | Uniform$(-1.5, -1.0)$ |

Concretely, for the representative system, the true enclosed area is $\mathbf{A_{true} = 21.602}$, while MetaSym predicts $\mathbf{A_{pred} = 21.619}$, corresponding to a relative deviation of *less than* $0.08\%$. This close agreement indicates that MetaSym not only reproduces the correct qualitative closed-orbit geometry (Fig. 7c), but also preserves phase-space volume to high accuracy during autoregressive rollouts. Such behavior is consistent with an approximately symplectic evolution and supports the claim that MetaSym captures fundamental geometric invariants of conservative dynamics beyond pointwise trajectory fitting.

## D.2 Empirical Perturbation Bound Estimation

The *ActiveDecoder* models the dissipative and control-input effects of a system as perturbations to the symplectic manifold that the conservative dynamics of a given system evolve on. While Section C.1 provides

a definitive argument in favor of this interpretation, calculating this bound analytically is not possible. To further support our claims we performed an ablation study that calculates this bound empirically, using the pretrained weights for a dissipative $3x3$ spring-mesh system with parameters such as decay that are chosen to represent a real deformable surface such as polypropylene undergoing extreme deformation (eg. a trampoline). In this edge case, The bound $C\rho$ is calculated as below:

$$\|d\Phi_{\theta^*}^\top \mathbf{J} d\Phi_{\theta^*} - \mathbf{J}\|_2 = 0.999 \tag{12}$$

where $\mathbf{J} = \begin{pmatrix} \mathbf{0} & \mathbf{I}_d \\ -\mathbf{I}_d & \mathbf{0} \end{pmatrix}$ and $\Phi_{\theta^*}$ the trained network.

### D.3 Effectiveness of Meta-Attention Mechanism

Our meta-learning framework adapts the Query and Value projections of the cross-attention in the ActiveDecoder, enabling task-specific modulation of decoded representations. As shown in Table 6, this meta-attention approach consistently outperforms MetaSym without meta-attention as well as standard pre-training and fine-tuning, across three dynamical systems with realistic dissipation and inertial parameters.

Table 6: Meta-learning performance (MSE $\pm$ std) across OOD systems.

|  | **3×3 Spring-mesh** | **Quantum System** | **Quadrotor** |
|---|---|---|---|
| **ActiveDecoder finetuning** | $0.405 \pm 0.536$ | $1.068 \pm 0.165$ | $8.814 \pm 3.328$ |
| **w/o meta-attention** | $0.613 \pm 0.432$ | $1.000 \pm 0.200$ | $16.457 \pm 15.184$ |
| **with meta-attention** | $\mathbf{0.230 \pm 0.115}$ | $\mathbf{0.898 \pm 0.241}$ | $\mathbf{8.397 \pm 5.906}$ |

Meta-learning shapes the parameter initialization such that gradients on a small adaptation set are more informative and better aligned with task-specific directions, enabling rapid specialization and improved training dynamics during adaptation. As a result, the model achieves lower MSE across all systems despite limited adaptation data, consistent with prior theoretical and empirical findings on gradient-based meta-learning in few-shot regimes (Raghu et al., 2020; Beck et al., 2025). In this ablation, we consider common nominal values for all system parameters in order to focus on the efficacy of the meta-learning mechanism rather than robustness to high-noise settings.

Finally, as showcased by Fig. 8 for the quadrotor, the fast-adaptation achieves smooth convergence as the overall training converges with significant loss minimization. This proves the smooth dynamics of our meta-learning framework and the absence of severe instabilities that could have hindered its performance.

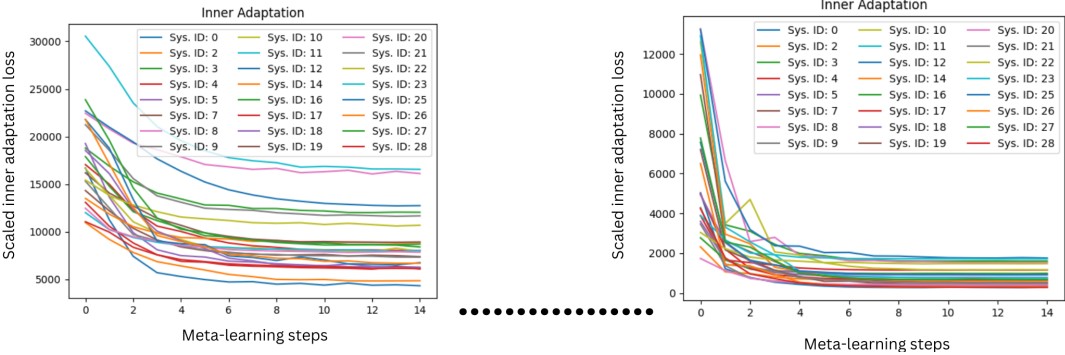

Figure 8: Inner Adaptation convergence for early training stage (left) and close-to-convergence training stage (right) for a quadrotor.

In Fig. 9, we also include heat maps that demonstrate the changes in queries and values due to the fine-tuning phase for two arbitrary system configurations.

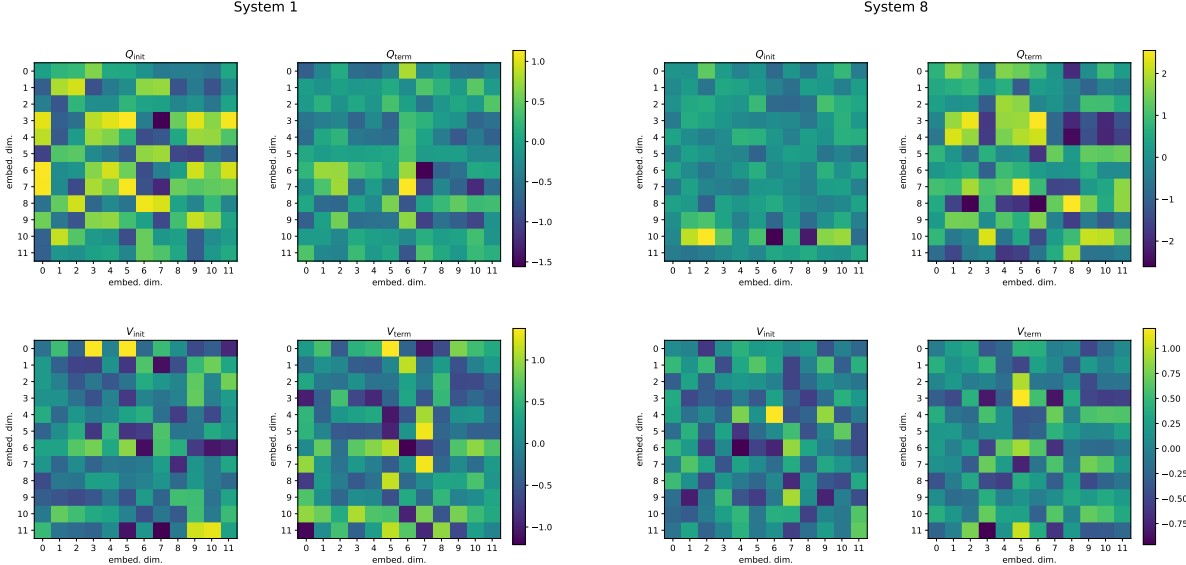

(a) Adaption of queries and values during fine-tuning     (b) Adaption of queries and values during fine-tuning

Figure 9: Adaption of queries and values during fine-tuning for two arbitrary systems.

## E   Context-Window Study

The context-window choice is application and task-dependent in general, since it comes as a trade-off between accuracy and long-term prediction. We provide in Table 7 a comprehensive study of different context-window lengths. In all of our results and studies we use a context-window of 30 time-steps, because we prioritize the long-term prediction capabilities of MetaSym.

Table 7: Mean Squared Error (MSE) with standard deviation ($\sigma$) for different context window sizes. As expected for Markovian dynamics, the error compounds as the context-window increases, however MetaSym retains a good trade-off between long-term prediction and accuracy.

| Context Window | MSE ($\pm\sigma$) |
|:---:|:---:|
| 2 | 0.2864 (0.2154) |
| 10 | 0.4482 (0.3134) |
| 20 | 0.9214 (0.8841) |
| 30 | 1.5903 (1.6514) |
| 50 | 3.5574 (3.8716) |
| 100 | 8.4024 (7.7815) |

## F   Compute Resources

The dataset for the $10 \times 10$ spring mesh system and the associated baselines and benchmarks were run on a Lambda cloud instance consisting of a NVIDIA A100 with 40GB of GPU memory and 200GB of RAM. All other experiments were run on a system containing 252GB of RAM, 32 core processor and a 24GB NVIDIA RTX 4090.

## G   Hyperparameters for Benchmarks

This section includes the hyper-parameters of our framework, which we used to benchmark each system as outlined in Table 1. This ensures the reproducibility and completeness of our method.

Table 8: Hyperparameter settings for MetaSym when benchmarking the spring-mesh.

| Hyperparameter | Encoder | Decoder |
|---|---|---|
| Optimizer | Outer: AdamW Inner: Adam | Outer: AdamW Inner: AdamW |
| Learning Rate | Outer: 0.001 Inner: 0.003 | Outer: 0.007 Inner: 0.01 |
| Inner Steps | 3 | 10 |
| Context Window | N/A | 30 |
| Dropout | 0.1$^*$ | 0.1 |
| Layers | 3 | 1 |
| Attention Heads | N/A | 4 |
| Early Stopping Epochs | 110 | 312 |
| Train / Val. dataset split [%] | 80 / 20 | 80 / 20 |
| Adapt / Infer. dataset split [%] | N/A | 30 / 70 |

$^*$DropConnect

Table 9: Hyperparameter settings for MetaSym when benchmarking the open-quantum system.

| Hyperparameter | Encoder | Decoder |
|---|---|---|
| Optimizer | Outer: AdamW Inner: Adam | Outer: AdamW Inner: AdamW |
| Learning Rate | Outer: 0.001 Inner: 0.003 | Outer: 0.008 Inner: 0.03 |
| Inner Steps | 3 | 15 |
| Context Window | N/A | 30 |
| Dropout | 0.2$^*$ | 0.2 |
| Layers | 3 | 1 |
| Attention Heads | N/A | 4 |
| Early Stopping Epochs | 294 | 354 |
| Train / Val. dataset split [%] | 80 / 20 | 80 / 20 |
| Adapt / Infer. dataset split [%] | N/A | 30 / 70 |

$^*$DropConnect

Table 10: Hyperparameter settings for MetaSym when benchmarking the quadrotor.

| Hyperparameter | Encoder | Decoder |
|---|---|---|
| Optimizer | Outer: AdamW Inner: Adam | Outer: AdamW Inner: AdamW |
| Learning Rate | Outer: 0.001 Inner: 0.003 | Outer: 0.01 Inner: 0.008 |
| Inner Steps | 3 | 15 |
| Context Window | N/A | 10 |
| Dropout | 0.4$^*$ | 0.45 |
| Layers | 3 | 1 |
| Attention Heads | N/A | 4 |
| Early Stopping Epochs | 34 | 222 |
| Train / Val. dataset split [%] | 80 / 20 | 80 / 20 |
| Adapt / Infer. dataset split [%] | N/A | 30 / 70 |

$^*$DropConnect

