# OpenReview forum: "MetaSym: A Symplectic Meta-learning Framework for Physical Intelligence"
_TMLR — Accepted by TMLR_

### Review · Reviewer_zQos · 2025-12-08

**Summary Of Contributions:**

Summary:
The paper introduces MetaSym, a meta-learning architecture for quick adaptation of dynamical systems across different environments and parameter settings. The framework combines a symplectic encoder for preserving shared geometric structure with an autoregressive decoder that adapts to non-conservative dynamics through a meta-attention mechanism. The encoder is built from SympNet blocks and trained bidirectionally to encourage Hamiltonian consistency and stable long-term rollouts and the decoder incorporates per-system parameters and control inputs. The approach is evaluated on a spring–mesh system, an open quantum system, and a quadrotor, showing improved prediction compared to selected baselines.

Strengths:
1. Architecture choices tied to physically meaningful design:
The model’s components correspond directly to distinct physical roles for example 1) a symplectic encoder built from canonical flow blocks to capture shared conservative structure, 2) an attention-based decoder to flexibly model dissipation, control inputs, and measurement effects, 3) the bidirectional (Δt/–Δt) training of the encoder to reflect time-reversibility in Hamiltonian systems.
2.  The method is tested on three very different domains: The paper evaluates the model on a mechanical system, an open quantum system, and a quadrotor sim-to-real task. This covers deterministic, stochastic, and controlled dynamics, which is an unusually broad range for a single framework. Showing good performance across all three suggests the approach is not tied to one specific type of system.
3. Clear separation of shared vs. task-specific dynamics: The model assigns the shared, conservative structure of the dynamics to the symplectic encoder, whose parameters are meta-learned and kept global across all systems. In contrast, variations that differ from system to system are handled by a small set of task-specific query/value parameters in the decoder that are adapted in the inner loop.

Weaknesses:
1. Unclear to what extent generalization is possible: Because the encoder is frozen after meta-training, it cannot adjust to new systems whose conservative structure meaningfully differs (for example different stiffnesses, masses, or Hamiltonian parameters). This makes it difficult to assess how far the model can generalize beyond the training distribution and raises questions about the strength of the fast adaptation claim.
2. The experiments do not show adaptation curves or comparisons with non-meta baselines under equal update budgets. In addition, the paper does not report any adaptation time, despite emphasizing “fast adaptation,” so it is unclear how efficient the inner-loop updates actually are in practice.
3. Structure-preservation is claimed but not evaluated: The symplectic encoder is justified by its ability to preserve geometric structure, but no explicit metrics are reported.
4.Limited baseline coverage: Although the paper discusses meta-learning methods such as iMODE in the related work, it does not include iMODE or similar task-adaptive baselines like CAVIA in the experimental comparison. These methods are designed for fast adaptation across dynamical systems and would provide a much clearer assessment of MetaSym’s meta-learning benefits.
5. Limited literature review: The paper emphasizes fast adaptation and the preservation of physical structure as core motivations, but the related work section does not discuss several relevant meta-learning approaches in scientific machine learning. For example, there is a substantial line of work on meta-learning physics-informed neural networks (meta-PINNs) and parameter-aware operator-learning models that pursue similar goals of learning shared physical structure and adapting quickly to new parameters.

**Audience:**

Yes

**Audience Explanation:**

The paper fits within scientific machine learning and topics like physics-informed learning, structure-preserving modeling, meta-learning for dynamics which are of interest to many researchers in this area. Its attempt to use a single architecture across a broad range of classical and  quantum dynamics and would likely attract attention even if some claims need refinement.

**Broader Impact Concerns:**

I do not see any ethical concerns.

**Claims And Evidence:**

No

**Claims Explanation:**

Several of the claims are not fully backed by the evidence shown. In particular, the meta-learning / “fast adaptation” benefits are not isolated via adaptation curves, controlled comparisons to non-meta baselines, so it is hard to attribute the gains specifically to the meta-learning design. The structure-preservation motivation (symplecticity, time reversibility) is also not evaluated with explicit geometric or physical metrics, despite being central to the framing. Finally, the claim of being the first physics-based model to adapt across classical and non-unitary quantum systems is not adequately justified in comparison to the related work. Please refer to the weaknesses for more information.

**Requested Changes:**

1) Please clarify and discuss the extend of generalization. Please clarify what forms of system variation the method is expected to handle (e.g., changes in conservative structure vs. non-conservative effects). Since the encoder is frozen at test time, it would be helpful to explicitly discuss this limitation and provide experiments or analysis showing how the method behaves when conservative parameters (e.g., masses, stiffnesses) shift outside the meta-training distribution.
2) Provide evidence isolating the meta-learning component. To support the “fast adaptation” claim, include adaptation curves or comparisons to non-meta baselines under equal update budgets.
3) Overall the results section needs to be explained with more details. For example, since the symplectic encoder is motivated by geometric and physical invariance, please report metrics that directly support these claims.
4) The experiments would be stronger with comparisons to more directly relevant meta-learning approaches (e.g., iMODE or CAVIA)
5) The related work section can be improved and expanded. For example by adding more directly relevant papers such as meta-learning PINNs.
6) Please report adaptation time.

---

> ### Author Response · Authors · 2025-12-25
> **Response to reviewer zQos & revised manuscript (clarification & additional results added)**
>
> We thank the reviewer for the detailed comments regarding generalization and meta-learning baselines.
>
> **"Unclear to what extent generalization is possible: Because the encoder is frozen... it cannot adjust to new systems whose conservative structure meaningfully differs."**
>
> * **Generalization Mechanism:** The SymplecticEncoder is meta-trained on a **distribution** of randomized conservative systems (varying masses, stiffnesses). This allows it to learn a generalized "basis" of canonical transformations. Then, the meta-learned ActiveDecoder adapts the system to varying dissipative/non-conservative phenomena and parameters. This acts as a perturbation to the SymplecticEncoder's latent symplectic manifold that we can naturally interpret as the effect of degeneracies caused by the varying parameters and non-conservative effects.
> * **OOD Performance:** Our experiments on the Quadrotor and Spring Mesh (**Tables 2b & 4b**) explicitly test on parameters (e.g., inertia, mass) significantly outside the training distribution. The low error confirms that the decoder successfully adapts the latent projections to handle these structural deviations.
>
> **"Limited baseline coverage... iMODE..."**
>
> We have added **iMODE** as a baseline for all three systems in **Table 1** (see below for a **partial view** / manuscript has been updated with the full benchmarking results). MetaSym consistently outperforms iMODE, particularly in structure preservation and error accumulation on the complex Quadrotor and Spring Mesh tasks. We would also like to point the reviewer to Appendix D.3, that includes the effectiveness of the meta-learning mechanism demonstrated by the reported MSE errors and  adaptation curves. We have now also included an ablation on fine-tuning vs meta-learning.
>
> **Table 1: Trajectory Error against Baselines (MSE )**
>
> | Models | Spring Mesh System | Quantum System | Quadrotor |
> | --- | --- | --- | --- |
> | **DHNNs** | 32.468 (28.086) | N/A | 26.375 (9.160) |
> | **Transformer** | 36.653 (15.618) | 0.950 (0.239) | 34.584 (14.063) |
> | **iMODE (New)** | **23.858 (11.552)** | **7.667 (0.436)** | **57.190 (20.903)** |
> | **MetaSym (Ours)** | **19.233 (15.673)** | **0.859 (0.215)** | **25.889 (8.967)** |
>
> **"Structure-preservation is claimed but not evaluated."**
>
> We have added extensive evaluations to support these claims:
>
> * **Physical Metrics:** In **Appendix D.1.2** (Harmonic Oscillator), we now report **Phase-Space Area conservation**, showing a relative deviation of less than **0.08%** from the true symplectic area.
> * **Empirical Symplectic Bound:** We point the reviewer to **Appendix D.2**, which reports the **Symplectic Jacobian Ablation**. We empirically calculated the bound  on the spring-mesh system and found it to be **0.999**, verifying the model stays close to the symplectic manifold. We provide the Jacobian ablation as **it is believed to be a more robust insight into symplecticity compared to phase space plots (Gruver et al., ICLR 2022; Hairer et al., 2006).**
>
> **"Limited literature review... Meta-PINNs."**
>
> We have expanded the **Related Work** to include **Meta-PINNs** and **Parameter-Aware Operator Learning**. We discuss how MetaSym differs: unlike PINNs which enforce physics via soft loss constraints (requiring known equations), MetaSym enforces geometric constraints architecturally, offering better stability for unknown or partially known dynamics, allowing purely data-driven training with structural ("hard") physics-informed characteristics.
>
> **"Please report adaptation time."**
> The adaptation and training wall-clock time is machine-dependent. It is also affected by the dimensionality of the problem. in **Appendix G** we report all the hyperparameters of the training procedure and **epochs for adaptation**.
>
> The inference time of the quadrotor (in non hard real-time settings) is **1.551 ms** per context window. Please refer to **Section 4.3** for further details.

---

> > ### Author Response · Authors · 2026-01-12
> >
> > Dear Reviewer zQos,
> >
> > As the final decision deadline is fast approaching this week, we would like to follow up on your feedback and our revised manuscript! Looking forward to your feedback and consideration.
> >
> > Best regards.

---

> > > ### Comment · Reviewer_zQos · 2026-01-20
> > > **Comment on the updates**
> > >
> > > The authors have answered my questions and I'm satisfied with the updates. No further comments.

---

> > > > ### Author Response · Authors · 2026-01-20
> > > >
> > > > Dear Reviewer zQos,
> > > >
> > > > We would like to thank the reviewer for their response and feedback. We are pleased to hear that their concerns have been successfully addressed.

---

### Review · Reviewer_bTsD · 2025-12-16

**Summary Of Contributions:**

The authors present MetaSym, a deep learning framework that employs meta-learning in conjunction with a symplectic encoder and an autregressive decoder. The idea of the former is to preserve the Hamiltonian, thereby allowing for true time-reversible dynamics. The authors benchmark this on 3 real-world datasets, demonstrating few-shot adaptation and lower errors than larger SoTA models.

**Additional Comments:**

I think the paper is reasonably well-written, with the exception of my concerns above.

**Audience:**

Yes

**Audience Explanation:**

I think this work is an important step in the right direction. The idea of combining meta-learning and symplecticity is quite timely, and has a wide variety of applications in classical and quantum mechanical systems, both static and dynamical.

**Broader Impact Concerns:**

None.

**Claims And Evidence:**

No

**Claims Explanation:**

I would say the results in this work support the claims, but that more evidence is needed. It isn't clear why the authors picked these particular benchmarks instead of somewhat obvious ones. For instance, examples from physics where analytical solutions are known would have been ideal, or at least cases where numerical solutions are available. When an architecture is presented and no connections are explicitly drawn between the architecture parameters and the generalization errors, very little can be gleaned from this work.

**Requested Changes:**

I request that the authors add the following experiments/details, in my view crucial for acceptance of this work:

1. A study of this approach on a baseline problem where an analytical solution is known, such as a harmonic oscillator or a more general N-body system such as the Calogero-Moser system.
2. For each test problem, a clear description of why symplecticity is required, and possibly some measure of the efficacy of the architecture in maintaining symplectity. If the architecture analytically enforces it, why the errors on the order of 10^{-2}? How much should one expect?
3. A clearer description of the symplectic architecture. What are alpha and beta? How are they chosen? What makes their particular network symplectic?
4. Many of the experiment descriptions are confusing. For instance, to me, quadrature is a numerical integration rule. It seems to mean something else in this context, or I'm not following why that's a quantity of interest. The descriptions of all experiments are somewhat niche and not clear at all to a broader audience. The authors should update these and take care to define terms before they're used.
5. The motivation for the chosen experiments isn't clear to someone like me who's not in this particular area. The authors could motivate things more clearly. If the experiments tell a story, what's the story? What other experiments might a broader audience who cares about Hamiltonians and symplecticity want to see?
6. Appendix C talks about departure from symplecticity to allow modeling of dissipative systems. It would be great to see what happens on varied problems without any dissipation, where one expects strict symplecticity and time reversibility.

---

> ### Author Response · Authors · 2025-12-25
> **Response to reviewer bTsD & revised manuscript (clarification & additional results added)**
>
> We appreciate the reviewer's feedback on the choice of benchmarks and the request for an analytical baseline.
>
> **"It isn't clear why the authors picked these particular benchmarks... examples from physics where analytical solutions are known would have been ideal."**
>
> * We selected the current benchmarks to stress-test high-dimensionality and real-world noise. More specifically:
>     * **Spring-mesh**: This is a 400-dimensional system with complex dynamics that can approximate phenomena in solid mechanics (see Section 4.1). This tests our method's scalability.
>     * **Open-Quantum System**: The system's inherent stochastic dynamics and exotic nature provides a canoncial, non-classical test-bed for our architecture. Its nature proves the generalization capabilities of MetaSym to non-classical mechanics.
>     * **Quadrotor**: The quadrotor provides a canonical, floating-base dynamics' example common in robotics. Testing MetaSym with real-world data proves our method's efficacy and robustness against real-world uncertainty.
>     * **Harmonic Oscillator:** Following the reviewer's suggestion, we trained MetaSym on a **Harmonic Oscillator** where the analytical solution is known. As detailed in **Appendix D.1.2**, MetaSym overlaps almost perfectly with the analytical solution, maintaining energy and phase-space volume conservation with high precision. We point the reviewer to **Appendix D.2**, which reports the **Symplectic Jacobian Ablation**. We empirically calculated the bound  on the spring-mesh system and found it to be **0.999**, verifying the model stays close to the symplectic manifold. We provide the Jacobian ablation as it is believed to be a more robust insight into symplecticity compared to phase space plots (Gruver et al., ICLR 2022; Hairer et al., 2006).
>
> **"If the architecture analytically enforces symplecticity, why the errors on the order of ?"**
>
> The **SymplecticEncoder** analytically enforces symplecticity. The reported bounded symplecticity error is induced by the **ActiveDecoder**, which *intentionally* perturbs the symplectic latent manifold to model the non-conservative effects (dissipation/control) and parameter variation. As seen in Appendix D.2, we bound this perturbation both empirically for a system with large amounts of dissipation by reporting the symplectic Jacobian of 0.999 and theortically as well. The reported MSE errors correspond to MetaSym’s autoregressive accuracy and numerical discretization errors and is distinct from the symplectic bound on the physics of the system. We have clarified this distinction and referenced the perturbation proof in **Appendix C**.
>
> **"A clearer description of the symplectic architecture. What are alpha and beta? How are they chosen?"**
>
> We have added the following clarification to **Section 3.1**:
>
> > "Here, we use $\alpha$ and $\beta$ that can be either linear or activation modules, **representing learnable parameters**, ensuring we remain in the class of canonical (i.e., symplectic) maps. This relies on the algebraic properties of the ''Up'' and ''Low'' updates, which act as symplectic shear transformations."
>
> **"Many of the experiment descriptions are confusing. For instance, to me, quadrature is a numerical integration rule..."**
>
> We have updated **Section 4.2** to explicitly define this term and included more explanations:
>
> > "In this context, **quadratures** (X) and (P) denote the non-commuting observables measured via heterodyne detection, serving as the quantum optical analogues to classical position and momentum (Milburn, 1987)."
>
> **"Appendix C talks about departure from symplecticity... It would be great to see what happens on varied problems without any dissipation."**
>
> We have now included an example of a simple harmonic oscillator. In **Appendix D.1.2** (Harmonic Oscillator), we now report **Phase-Space Area conservation**, showing a relative deviation of less than **0.08%** from the true symplectic area. We also plot **energy drift** and **trajectory evolution**.

---

> > ### Author Response · Authors · 2026-01-12
> >
> > Dear Reviewer bTsD,
> >
> > As the final decision deadline is fast approaching this week, we would like to follow up on your feedback and our revised manuscript! Looking forward to your feedback and consideration.
> >
> > Best regards.

---

> > > ### Comment · Reviewer_bTsD · 2026-01-13
> > >
> > > I'm happy with the updates.

---

> > > > ### Author Response · Authors · 2026-01-13
> > > >
> > > > Dear Reviewer bTsD,
> > > >
> > > > We would like to thank the reviewer for their response and previous insights. We are glad to have addressed their concerns.

---

### Review · Reviewer_7FVn · 2025-12-17

**Summary Of Contributions:**

The paper proposes MetaSym, a two-part architecture for learning physical dynamics:

(i) a SymplecticEncoder (a stack of SympNet blocks trained forward/backward to enforce time-reversal and preserve geometric invariants),

(ii) an ActiveDecoder (autoregressive transformer with a meta-attention head; only query/value are adapted per system for few-shot transfer).

The system is evaluated on (i) a 400-D spring-mesh, (ii) an open quantum heterodyne-measurement task, and (iii) a quadrotor (sim→real fine-tuning). Table 1 (p. 9) reports lower MSE with fewer parameters than Transformers, DHNN/SymODEN, and FNO on these tasks; figures on pp. 7–9 illustrate long-horizon rollouts and real-world quadrotor predictions.

Strengths:

Building on SympNets gives a clean way to encode canonical maps and time-reversibility; the bidirectional training is well motivated and clearly explained.

The datasets span classical, stochastic/open-quantum, and robotics domains, including real-world quadrotor adaptation with tight inference time.

The meta attention layer is simple and practically well motivated.

Weaknesses:

While the encoder is symplectic, the decoder perturbs the latent to model non-conservative effects. As written, it’s unclear how well the overall map preserves invariants over long horizons or what is guaranteed beyond “near-symplectic.”  Physics metrics (energy drift, momentum error, phase-space volume change) were not reported to fully understand the capability of the model.

The paper says ablations are in App. D, but the main text doesn’t isolate: (i) plain transformer decoder with vs without the SymplecticEncoder; (ii) symplectic pre-training vs random init; (iii) meta-attention (Q/V adaptation) vs full fine-tuning vs none. A small table/plot in the main paper would make the causal story crisp.

The quantum task is intrinsically noisy, however, any confidence interval (CI) plot is not provided.

**Audience:**

Yes

**Audience Explanation:**

It is useful for the community.

**Claims And Evidence:**

No

**Claims Explanation:**

The paper is clearly written. The motivations and the claimed contributions are solid. There are some concerns regarding the symplectic structure claim, as the empirical results do not evaluate that. If the paper shows the evaluation, I would be happy to accept.

**Requested Changes:**

Please quantify with physics metrics (energy drift, momentum error, phase-space volume change) to understand whether the model is truly symplectic. Also, is there any theoretical guarantee on retaining symplectic structure?

The reviewer would like to see some ablation studies, as pointed out in the weakness.

 Please specify training split, adaptation shots/steps, units, and scaling of each channel, and whether the reported error is averaged across states or weighted. Also, add a runtime breakdown (encoder/decoder) to support the real-time claim, especially for the quadrotor experiment.

What breaks if the context window is mis-specified (too short/long)?

---

> ### Author Response · Authors · 2025-12-25
> **Response to reviewer 7FVn & revised manuscript (clarification & additional results added)**
>
> We thank the reviewer for the positive assessment of the paper's clarity and solid contributions. We have addressed the specific questions regarding symplectic structure preservation and ablations below.
>
> **"As written, it’s unclear how well the overall map preserves invariants over long horizons or what is guaranteed beyond “near-symplectic.” Physics metrics... were not reported"**
>
> We agree that quantifying this is essential. We clarify that the **SymplecticEncoder** is strictly symplectic by design (preserving invariants), while the **ActiveDecoder** applies perturbations to model non-conservative effects (dissipation/control) and parameter variations.
>
> * **Theoretical Guarantee:** We direct the reviewer to **Appendix C (Theorem C.1)**, where we prove that the model is **-near-symplectic**. The deviation from symplecticity is strictly bounded by the magnitude of the non-conservative effects.
> * **Empirical Metrics:** We point the reviewer to **Appendix D.2**, which reports the **Symplectic Jacobian Ablation**. We empirically calculated the bound  on the spring-mesh system and found it to be **0.999**, verifying the model stays close to the symplectic manifold. We provide the Jacobian ablation as it is believed to be a more robust insight into symplecticity compared to phase space plots (Gruver et al., ICLR 2022, arXiv:2202.04836; Hairer et al., 2006).
> * **Physics Metrics:** We added **Appendix D.1.2**, benchmarking a conservative **Harmonic Oscillator**. As shown in **Figure 7**, MetaSym achieves a relative deviation of less than **0.08%** from the true symplectic phase-space area ( vs ), demonstrating minimal energy drift.
>
> **"The paper says ablations are in App. D, but the main text doesn’t isolate: (i) plain transformer decoder... (ii) symplectic pre-training... (iii) meta-attention vs full fine-tuning."**
>
> * **Transformer:** This baseline is provided in **Table 1**. The "Transformer" baseline represents our decoder architecture trained end-to-end without the SymplecticEncoder but with a Transformer encoder instead. MetaSym outperforms it significantly (e.g., **19.23 vs 36.65 MSE** on Spring Mesh).
> * **Symplectic Pre-training:** Training the SympNet from "random initialization" without conservative data is physically invalid, as the network must learn a basis of canonical transformations to function as a symplectic encoder. Hence, the SymplecticEncoder's training data must be conservative.
> * **Meta-attention vs. Full Fine-tuning:** We have added **Appendix D.3** and **Table 6** (below) to explicitly compare **Meta-Attention** (adapting only Q/V) against **Full Fine-Tuning** of the ActiveDecoder. The results demonstrate that adapting only the attention parameters provides superior generalization by avoiding overfitting in the few-shot regime. This advantage arises because meta-learning gradients are inherently more informative. Encoding task-agnostic update directions, shaped across a distribution of tasks, promotes robust adaptation under limited data and is consistent with theoretical and empirical analyses (Please refer to Appendix D.3).
>
> | Method                                   | 3×3 Spring-Mesh | Quantum System | Quadrotor |
> |------------------------------------------|-----------------|----------------|-----------|
> | MSE with ActiveDecoder finetuning (±σ)   | 0.405 (0.536)   | 1.068 (0.165)  | 8.814 (3.328) |
> | MSE w/o Meta-Attention (±σ)              | 0.613 (0.432)   | 1.000 (0.200)  | 16.457 (15.184) |
> | MSE with Meta-Attention (±σ)             | **0.230 (0.115)** | **0.898 (0.241)** | **8.397 (5.906)** |
>
>
> **"The quantum task is intrinsically noisy, however, any confidence interval (CI) plot is not provided"**
>
> In **Table 1** we include the standard deviation for all trajectory MSEs. Additionally, we point out that the Wiener increment from the measurement backaction is stochastic and obtained from a Gaussian distribution. We have included a visualization of the confidence intervals in **Figure 5** of the Appendix B.2.
>
> **"Please specify training split, adaptation shots/steps, units, and scaling... Also, add a runtime breakdown..."**
>
> We have added **Appendix G** and expanded **Appendix B** to specify the exact training splits, adaptation steps, and parameter units. Regarding runtime, we performed an analysis on the Quadrotor task (NVIDIA RTX 4090). As noted in **Section 4.3**:
>
> > "The results indicate that MetaSym can run on an NVIDIA RTX4090 in **1.551 ms** for its full 30-timestep context-window."
>
> **"What breaks if the context window is mis-specified (too short/long)?"**
>
> The manuscript includes a **Context Window Study** in **Appendix E** (Table 7). As expected for Markovian dynamics, the error compounds as the context-window increases, however, MetaSym retains a good trade-off at 30 steps.
>
> | Context Window | MSE (std.) |
> | --- | --- |
> | 2 | 0.2864 (0.2154) |
> | 10 | 0.4482 (0.3134) |
> | **30 (Selected)** | **1.5903 (1.6514)** |
> | 100 | 8.4024 (7.7815) |

---

> > ### Author Response · Authors · 2026-01-12
> >
> > Dear Reviewer 7FVn,
> >
> > As the final decision deadline is fast approaching this week, we would like to follow up on your feedback and our revised manuscript! Looking forward to your feedback and consideration.
> >
> > Best regards.

---

> > > ### Comment · Reviewer_7FVn · 2026-01-13
> > > **Thank you**
> > >
> > > I would like to thank the authors for clarifying my doubts. I have one comment, for which the authors should provide some discussion in the final version. The need for conservative data pretraining of the SymplecticEncoder is reasonable, but the paper should state limitations if conservative data are scarce and clarify how sensitive results are to encoder mis-specification

---

> > > > ### Author Response · Authors · 2026-01-13
> > > >
> > > > Dear Reviewer 7FVn,
> > > >
> > > >
> > > > We thank the reviewer for their response and positive comments.
> > > >
> > > > The requirement for conservative data during pretraining of the SymplecticEncoder arises from its role in learning a canonical latent representation that enforces hard symplectic constraints. When conservative data are available but limited, our results indicate that MetaSym remains robust: as shown in Table 1, MetaSym achieves strong performance with fewer parameters than black-box baselines. This is also consistent with prior observations from Jin *et.al* on SympNets, where their strong geometric inductive biases substantially reduce sample complexity and long horizon autoregressive behavior (40 timesteps training / 100 autoregressive inference for simple conservative systems).
> > > >
> > > > In the regime where conservative data are severely limited or entirely unavailable, the encoder can no longer be expected to reliably recover the true canonical structure. In this case, we expect the black-box nature and expressive capacity of the ActiveDecoder to partially compensate for encoder mis-specification. This expectation is motivated by our transformer benchmark in Table 1, which demonstrates that an unconstrained autoregressive decoder trained purely on non-conservative data can achieve reasonable predictive accuracy, albeit without structural guarantees. MetaSym interpolates between these regimes: as conservative SymplecticEncoder pretraining becomes extremely small or non-existent, the contribution of the symplectic inductive bias weakens and the overall behavior approaches that of a black-box autoregressive model. However, we emphasize that in this setting there are no guarantees on invariant preservation or long-horizon stability, and the degree of compensation is expected to be system- and data-dependent.
> > > >
> > > > Following your suggestion, we will explicitly state this limitation in the final manuscript. In particular, we will clarify that MetaSym’s strongest guarantees (e.g. bounded energy drift, near-symplectic behavior, and long-horizon stability) rely on the availability of at least some representative conservative data for pretraining the SymplecticEncoder. When such data are unavailable, MetaSym can be interpreted as a hybrid physics-informed / data-driven model whose guarantees degrade continuously toward those of a fully black-box architecture.
> > > >
> > > > We thank the reviewer for prompting this clarification, which improves the scope and transparency of the method.

---

### Decision · Action_Editor_c9QT · 2026-01-30

**Recommendation:** Accept as is

**Audience:**

Yes

**Audience Explanation:**

The paper targets an active area in scientific ML (structure-preserving learning + meta-learning) and evaluates on diverse and realistic settings (high-dimensional mechanics, open quantum dynamics, and sim-to-real quadrotor). The method and results should be useful to readers working on physics-informed learning, dynamical systems, robotics, and beyond.

**Claims And Evidence:**

Yes

**Claims Explanation:**

This paper claims that MetaSym can learn and adapt physical dynamics efficiently by combining a symplectic encoder (to preserve structure) with a meta-learning autoregressive decoder (to handle system variation and non-conservative effects). The original submission already supported these claims with strong results on three diverse and realistic benchmarks (spring-mesh, open quantum dynamics, and sim-to-real quadrotor), showing improved accuracy and few-shot adaptation compared to competitive baselines. The rebuttal and revision further strengthened the evidence by adding explicit structure-preservation evaluations (harmonic oscillator with very small energy/phase-space area deviation and a symplectic Jacobian check), expanding baseline coverage (including iMODE), and providing clear ablations that isolate the benefit of meta-attention adaptation over full fine-tuning.